# AHA - Predicting What Matters Next: Online Highlight Detection Without Looking Ahead

**Aiden Chang**[*]
University of Southern California
Los Angeles, CA 90089
aidenchang@gmail.com

**Celso De Melo**
DEVCOM Army Research Laboratory
Adelphi, MD 20783

**Stephanie M. Lukin**
DEVCOM Army Research Laboratory
Adelphi, MD 20783

## Abstract

Real-time understanding of continuous video streams is essential for intelligent agents operating in high-stakes environments, including autonomous vehicles, surveillance drones, and disaster response robots. Yet, most existing video understanding and highlight detection methods assume access to the entire video during inference, making them unsuitable for online or streaming scenarios. In particular, current models optimize for offline summarization, failing to support step-by-step reasoning needed for real-time decision-making. We introduce **AHA**, an autoregressive highlight detection framework that predicts the relevance of each video frame against a task described in natural language. Without accessing future video frames, AHA utilizes a multimodal vision-language model and lightweight, decoupled heads trained on a large, curated dataset of human-centric video labels. To enable scalability, we introduce the Dynamic SinkCache mechanism that achieves constant memory usage across infinite-length streams without degrading performance on standard benchmarks. This encourages the hidden representation to capture high-level task objectives, enabling effective frame-level rankings for *informativeness*, *relevance*, and *uncertainty* with respect to the natural language task. AHA achieves state-of-the-art (SOTA) performance on highlight detection benchmarks, surpassing even prior offline, full-context approaches and video-language models by +5.9% on TVSum and +8.3% on Mr.Hisum in mAP (mean Average Precision). We explore AHA's potential for real-world robotics applications given a task-oriented natural language input and a continuous, robot-centric video. Both experiments demonstrate AHA's potential effectiveness as a real-time reasoning module for downstream planning and long-horizon understanding.

## 1 Introduction

Real-time understanding of continuous video streams is crucial for intelligent agents operating in high-stakes environments, from autonomous vehicles and surveillance drones to field-deployed robotics in disaster relief scenarios [1–4]. Despite this need, while earlier explorations into Online Highlight Detection (OHD) existed (e.g., using LSTMs [5]), the trajectory of contemporary HD research, particularly leveraging powerful modern transformer based architectures, has overwhelmingly centered on offline, full-context processing [6–8]. Even approaches incorporating task-conditioning via

---

[*]Conducted research as a fellow at the DEVCOM Army Research Laboratory.
[1]github.com/aiden200/Aha-

39th Conference on Neural Information Processing Systems (NeurIPS 2025).

natural language queries operate offline assume the entire video is available during inference [9, 10]. This fundamental reliance on full-context renders existing HD methods unsuitable for streaming applications requiring step-by-step reasoning and immediate action based on unfolding events.

Concurrently, a separate area of research has explored streaming video analysis, often leveraging Large Language Models (Video-LLMs) for tasks like dense video captioning or generating dialogue responses about ongoing events [11, 12]. While some of these models have explored HD as an auxiliary capability, their application to OHD faces significant limitations. These Video-LLMs often necessitate modifications to standard HD benchmarks, employ post-hoc smoothing techniques that violate strict online constraints by implicitly using future information, and ultimately yield suboptimal HD performance [13]. This leaves a critical gap: a robust method designed specifically for accurate, online, task-conditioned highlight detection on standard benchmarks.

We address this gap by introducing a novel framework built for OHD. We define OHD as the method of analyzing a streaming video by observing frames strictly one at a time and, for each current frame, predicting its highlight score using only past and present information, without accessing any future frames. This sequential, causal processing is fundamental for enabling real-time decision-making in dynamic environments. Given a natural language task description, our model, AHA, performs OHD by employing a lightweight, autoregressive scoring mechanism focused directly on highlight detection. This allows AHA to operate effectively on traditional HD benchmarks in a truly online fashion, without requiring benchmark modifications or non-causal smoothing, and achieving SOTA performance even in zero-shot settings. Our main contributions are:

**AHA Framework for Efficient OHD:** We propose AHA, an autoregressive framework featuring lightweight prediction heads (scoring relevance, informativeness, uncertainty) and our novel Dynamic SinkCache memory for efficient, constant-cost, OHD under natural language conditioning, and a video quality dropout mechanism to enhance robustness against real-world noise.

**A Large-Scale Dataset for OHD** [2]**:** We construct and release the Human Intuition Highlight Dataset (HIHD), a novel dataset of ~23k videos incorporating user engagement signals and task-driven captions, specifically designed to train and benchmark task-conditioned OHD models.

**SOTA OHD:** AHA surpasses prior methods, including offline approaches, on the HD benchmarks TVSum [14] (+5.9% mAP) and Mr.Hisum [15] (+8.3% mAP). We validate AHA's robustness and real-world applicability through comprehensive experiments, ablations, and on a challenging long-horizon, noisy robotics video from SCOUT [16], demonstrating task-relevant understanding where offline processing is infeasible.

## 2    Related Works

**Offline and OHD.** HD research, especially with modern architectures, has predominantly focused on offline, full-context processing. Techniques evolved from early handcrafted features to deep attention models [17–19, 10], but fundamentally require offline access. These methods, while achieving strong offline results, require bidirectional temporal access, making them unsuited for streaming. To the best of our knowledge, one of the few recent attempts at dedicated OHD using sequential models was Lal et al. [5] with LSTMs; yet, the challenge of frame-wise highlight prediction under strict online causality remains mostly open.

A central and persistent challenge in HD is the difficulty of obtaining labels that accurately reflect what constitutes a highlight, and, critically, generalizing this understanding across diverse video domains and content types [20, 15]. While early benchmarks like TVSum [14] offered rich but small-scale human annotations (50 videos), later datasets like Mr.Hisum [15] leveraged large-scale user engagement signals (e.g., replay spikes) for scalability and capturing broader interest. Their underlying hypothesis, which we also explore, is that these large-scale engagement patterns effectively capture moments of high viewer interest that align with highlight-worthy content, which correspond to human intuition. While existing datasets capture this intuition, robust OHD for diverse, raw streaming is often hindered by limitations such as the small-scale of benchmarks (e.g., TVSum) or, even in larger collections, by 'clearer' signals and content less representative of the variable quality and uncurated nature of many live streams. Building on prior theory to address these gaps, we introduce a new large-scale dataset that utilizes engagement-style signals akin to these datasets about

---

[2]The instructions on how to download the dataset will be included in the github.

human intuition, but is distinctively curated for broader visual quality variance and explicit support for task-conditioned learning.

**Streaming Video-Language Models.** Recent Streaming Video-Language Models (Video-LLMs) [10, 12, 11, 21] have significantly advanced multimodal reasoning for streaming video, offering architectural inspirations, particularly in memory-efficient processing (e.g., StreamingLLM [21], Token Turing Machines [22]), which inform our work. However, their direct application to robust OHD reveals limitations. These models often prioritize interactive tasks (e.g., dialogue, VQA) over the continuous, fine-grained scoring essential for OHD.

Consequently, when highlight detection is addressed, it is often as an auxiliary function, with modified evaluation criteria that is not aligned with traditional HD benchmarks or strict online constraints, leading to performance that can be suboptimal for specialized HD [13]. Moreover, their evaluations often use short clips, rarely demonstrating sustained OHD performance on long videos (>10 minutes).

AHA diverges from this trend by being specifically architected for efficient, high-performance OHD. It adapts memory-efficient streaming concepts for the video-language domain by employing task-aware scoring heads that process video as a continuous stream against a persistent task. Crucially, AHA demonstrates its effectiveness not only by outperforming traditional highlight detection benchmarks under strict online settings but also by maintaining robust performance on long-form videos, addressing a key gap in current streaming literature.

**Uncertainty-Aware Online Modeling** Unlike HD models offering deterministic scores, AHA incorporates an uncertainty head drawing from probabilistic sequence learning [23–25] to model predictive uncertainty, crucial for online settings with limited context. To our knowledge, AHA is the first application of explicit probabilistic uncertainty modeling for task-conditioned, frame-level OHD.

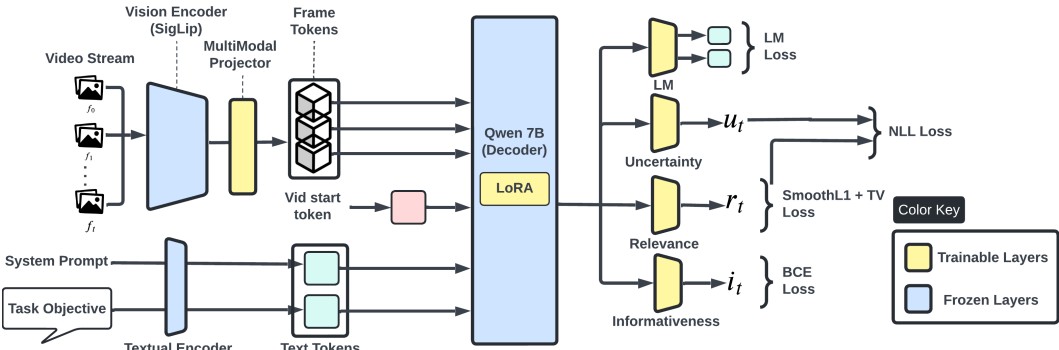

Figure 1: AHA architecture, showing the flow from video stream and text prompts through the visual encoder, multimodal projector, decoder, to the multi-objective prediction heads.

## 3 Methodology

We treat task-conditioned OHD in streaming video as follows: given a continuous stream of video frames $\{f_0, f_1, \ldots, f_t\}$, the goal at each timestep $t$ is to predict a scalar highlight score $\hat{y}_t$ for the current frame $f_t$, indicating its relevance to a user-specified natural language task objective. This objective, $\mathcal{Q} = \{q_1, \ldots, q_k\}$, and an optional system prompt $\mathcal{S} = \{s_1, \ldots, s_m\}$ are provided once at the beginning of the video and remain fixed throughout inference. At each inference timestep, the model observes (1) the current frame $f_t$, (2) the fixed task and system prompt embeddings $\mathcal{Q}$ and $\mathcal{S}$, and (3) a memory mechanism (e.g., a KV cache [26]) that stores previously computed tokens for efficient autoregressive decoding. The model must emit a scalar highlight score $\hat{y}_t$ without access to future frames, full-sequence context, or bidirectional attention, enabling real-time, low-latency operation. During training, instead of a KV cache, the model uses a fixed-length window of preceding frame and text tokens. This constrained context ensures the model learns to operate under streaming conditions while still allowing for full gradient flow. Auxiliary objectives (e.g., language modeling or captioning) may be incorporated to enhance semantic representations, but the training regime mirrors the **causal constraint:** the model sees only the current and past tokens within a limited window.

To solve this OHD problem, we propose AHA, a lightweight autoregressive framework built on recent advances in streaming multimodal LLMs. Its architecture (Fig. 1) consists of four key components: (1) A **Frozen Visual Encoder** (pretrained SigLIP [27]) extracts frame features, facilitating generalization without visual fine-tuning [13, 12]. (2) A **Minimal Multimodal Projection**, a single linear layer that maps visual embeddings to the LLM token space for fast per-frame tokenization. (3) A **Token-Level Autoregressive Decoder**, a decoder-only transformer [28, 29], processes interleaved text (including $\mathcal{S}$ and $\mathcal{Q}$ initially) and visual tokens in a unified sequence, enabling continuous, single-pass decoding for streaming inference. (4) **Multi-Objective Prediction Heads** (relevance, informativeness [13], uncertainty) are added on top of the decoder's final hidden layer $h_t$ to capture frame-level semantics for HD, which are then combined to produce $\hat{y}_t$. An auxiliary language modeling head (LM head) [12] also enriches representations during training. The selection of SigLIP and the Qwen2-based decoder is grounded in their SOTA performance and widespread adoption in recent vision-language literature (see Appendix C.5).

## 3.1 Training Objectives

We supervise AHA by jointly training four lightweight prediction heads, each targeting a distinct objective: task-conditioned relevance, informativeness, uncertainty, and auxiliary captioning. The total loss is a fixed, weighted sum of these objectives, a simple yet theoretically grounded multi-task learning strategy (see Appendix C.5 for a detailed justification of this approach and our model backbone selection).

**Relevance Head.** The relevance head estimates task-conditioned highlight relevance for frame $f_t$ via a scalar prediction $\hat{r}_t = W_r h_t$, where $h_t \in \mathbb{R}^{D_{hidden}}$ is the decoder's final hidden state and $W_r \in \mathbb{R}^{D_{hidden} \times 1}$ are learned linear projection weights. This prediction is supervised against human engagement scores $r_t$ (Sec. 3.3) using a Smooth L1 loss [30], $\mathcal{L}_{\text{relevance}}$ (Eq. 1a). To encourage smooth temporal predictions reflecting common user engagement patterns often observed in video data (e.g., gradual build-up and fall-off of interest around key moments, as noted in datasets like Mr.Hisum), we incorporate an additional total variation (TV) regularizer [31], $\mathcal{L}_{\text{TV}}$ (Eq. 1b). These are given by:

$$\mathcal{L}_{\text{relevance}} = \text{SmoothL1}(\hat{r}_t, r_t) \quad \text{(1a)} \qquad \mathcal{L}_{\text{TV}} = \frac{1}{T_{win} - 1} \sum_{t=1}^{T_{win}-1} v_t (\hat{r}_t - \hat{r}_{t-1})^2 \quad \text{(1b)}$$

where $v_t \in \{0, 1\}$ in Eq. (1b) indicates if both $\hat{r}_t$ and $\hat{r}_{t-1}$ are valid within a temporal window of size $T_{win}$. The total relevance objective, $\mathcal{L}_{\text{relevance-total}}$, combines these terms:

$$\mathcal{L}_{\text{relevance-total}} = \mathcal{L}_{\text{relevance}} + \lambda_{\text{TV}} \mathcal{L}_{\text{TV}} \quad (2)$$

**Informativeness Head.** Informativeness measures whether a frame introduces new information relative to recent context. Following prior work in dialog-based VideoLLMs [13], we incorporate a binary classification head to estimate if frame $f_t$ introduces new information, outputting a score $\hat{i}_t = \text{softmax}(W_i h_t)$, where $W_i \in \mathbb{R}^{D_{hidden} \times 2}$ are learned weights projecting $h_t$ to a 2D output for binary classification. It is trained to recognize temporally novel or redundant frames using Binary Cross-Entropy (BCE) with ground truth $i_t \in \{0, 1\}$ (Sec. 3.3):

$$\mathcal{L}_{\text{info}} = \text{BCE}(\hat{i}_t, i_t) \quad (3)$$

**Uncertainty Head.** Uncertainty captures the model's confidence in its frame-level predictions under partial observability. This head predicts the logarithm of Gaussian variance. From the hidden state $h_t$, its linear projection $W_u$ outputs a raw log-variance $\hat{l}_t = W_u h_t$. This is clamped to a predefined range $[L_{min}, L_{max}]$ (yielding $\hat{l}_{t,c}$) to obtain the predicted variance $\hat{\sigma}_t^2 = \exp(\hat{l}_{t,c})$. The primary loss component is the Gaussian negative log-likelihood (NLL) [25]. For this, the mean $\mu_t$ is taken as the linear output of the relevance head (i.e., $\mu_t = \hat{r}_t$), used alongside the ground truth relevance $r_t$ and the predicted variance $\hat{\sigma}_t^2$:

$$\mathcal{L}_{\text{NLL}} = \frac{(r_t - \mu_t)^2}{2\hat{\sigma}_t^2 + \delta} + \frac{1}{2} \log(2\pi \hat{\sigma}_t^2 + \delta) \quad (4)$$

where $\delta$ is a small stability constant (e.g., $10^{-6}$). However, relying solely on the NLL loss can lead to mode collapse: a known pitfall where the model learns a degenerate solution by predicting a single, uninformatively high variance for all frames to trivially minimize the loss [32]. To counteract this, we introduce a variance diversity penalty, $\mathcal{L}_{\text{div}}$ (Eq. 5a), based on the batch standard deviation of the

clamped log-variances. This regularizer forces the model to produce a dynamic and meaningful range of uncertainty values. The final uncertainty loss, $\mathcal{L}_{\text{uncertainty}}$ (Eq. 5b), combines the expected NLL with this penalty. A detailed derivation and justification for this uncertainty formulation is provided in Appendix C.3.

$$\mathcal{L}_{\text{div}} = -\lambda_{div} \cdot \text{std}(\{\hat{l}_{i,c}\}_{i \in \text{batch}}) \quad (5a) \qquad \mathcal{L}_{\text{uncertainty}} = \max(0, \mathbb{E}[\mathcal{L}_{\text{NLL}}] + \mathcal{L}_{\text{div}}) \quad (5b)$$

**LM Head.** Following prior streaming LLMs [13, 12], an auxiliary LM head encourages semantically rich hidden representations. At randomly sampled training timesteps, the model generates short captions (Sec. 3.3) for the current frame conditioned on prior context using standard cross-entropy loss for next token prediction:

$$\mathcal{L}_{\text{LM}} = \text{CrossEntropy}(\text{LMHead}(h_t), y_t) \qquad (6)$$

Generated text is not injected back into the context, focusing on unidirectional frame-wise scoring.

**Total Loss.** The final training objective, $\mathcal{L}_{\text{total}}$, is a weighted combination of the $\mathcal{L}_{\text{relevance-total}}$ (Eq. 2), $\mathcal{L}_{\text{info}}$ (Eq. 3), $\mathcal{L}_{\text{uncertainty}}$ (Eq. 5b), and $\mathcal{L}_{\text{LM}}$ (Eq. 6):

$$\mathcal{L}_{\text{total}} = \lambda_r \mathcal{L}_{\text{relevance-total}} + \lambda_i \mathcal{L}_{\text{info}} + \lambda_u \mathcal{L}_{\text{uncertainty}} + \lambda_{\text{LM}} \mathcal{L}_{\text{LM}} \qquad (7)$$

Fixed weights $\lambda$ are used during training (see Appendix C.2 for values). We use fixed weights to ensure training stability and interpretability, avoiding the complexity of joint optimization or dynamic reweighting across heterogeneous objectives. Additional implementation details and motivation for each loss component are provided in Appendix C.1.

## 3.2 Inference and Memory Management

Transformer self-attention scales quadratically with sequence length [33], making streaming inference costly. To mitigate this, we adopt a KV Cache [26], storing previously computed attention keys and values at each layer, avoiding redundant computation. Each layer's cache grows with sequence length $L$, storing tensors of shape $[B, H, L, D]$, where $B$ is batch size, $H$ is number of heads, and $D$ is head dimension. However, for long videos ($L_{\text{max}} > 127\text{k}$ in our evaluations), this unbounded growth leads to GPU out-of-memory (OOM) errors, highlighting the need for a memory-efficient alternative.

**Dynamic SinkCache.** To solve this, our framework introduces the Dynamic SinkCache, a novel modification of the hybrid memory approach from SinkCache [21]. Unlike the standard method that uses the first few generic tokens as its sink, our mechanism creates a more targeted long-term memory. It dynamically constructs the sink to contain exclusively the natural language task objective tokens ($\mathcal{Q}$), while the sliding window is dedicated to recent visual context. This design carries a constant memory footprint, supporting inference over arbitrarily long videos. In our implementation, the task objective sink averages ~45 tokens, which we pair with a sliding window of 2048 recent tokens. This configuration, requires only 17% of the standard cache ($L_{\text{avg}} = 12{,}421$) and conserves memory while achieving improved performance (see Section 4.2).

Formally, the highlight score at timestep $t$ is computed as $\hat{y}_t = f_\theta(f_t, \mathcal{Q}, \mathcal{S}, \mathcal{K}_t)$. The term $\mathcal{K}_t = \{\mathcal{Q}, k_{t-n:t}\}$ represents the memory accessible at timestep $t$. Here, the sink is precisely the set of task objective tokens $\mathcal{Q}$, and $k_{t-n:t}$ is the sliding window of $n$ recent visual tokens.

A comprehensive explanation of the Dynamic SinkCache mechanism, including its operational details, comparisons against other caching mechanisms, the role of sink tokens in maintaining long-term context, and an illustrative diagram, is provided in Appendix F.

**Scoring.** The final highlight score $\hat{y}_t$ is computed by fusing the relevance ($\hat{r}_t$), informativeness ($\hat{i}_t$), and uncertainty ($\hat{u}_t$) heads using an uncertainty-aware, piecewise scoring function. Specifically, let $\hat{r}_t$ be the predicted relevance ($W_r h_t$), $\hat{i}_t$ the predicted informativeness (softmax($W_i h_t$)), and $\hat{u}_t$ the predicted uncertainty score (taken as the clamped log-variance $\hat{l}_{t,c}$ output by the uncertainty head, where $\hat{l}_{t,c} = \text{clamp}(W_u h_t, L_{min}, L_{max})$. We then apply an uncertainty-aware linear weighting function:

$$\hat{y}_t = \begin{cases} \alpha \hat{i}_t + \beta \hat{r}_t, & \text{if } \hat{u}_t \leq \tau_u \quad \text{(low uncertainty)} \\ \alpha \hat{i}_t + \beta \hat{r}_t - \epsilon(\hat{u}_t - \tau_u), & \text{if } \hat{u}_t > \tau_u \quad \text{(high uncertainty)} \end{cases} \qquad (8)$$

The parameters $(\alpha, \beta, \epsilon, \tau_u)$ are set using a static approach, which our analysis shows is more robust than unstable dynamic alternatives (see Appendix C.4). This framework offers a flexible trade-off:

for a **truly zero-shot** configuration, we use a fixed heuristic (based on a 10:7 ratio for $\alpha : \beta$). For **optimal domain-adapted** performance, all four parameters are tuned via a lightweight grid search. Our results in Section 4.1 are presented for both settings to demonstrate the model's capabilities.

### 3.3 Video Datasets for Prediction Head Supervision

We train AHA using a combination of existing video-language datasets and a novel dataset tailored for highlight relevance in videos. These datasets supervise different model heads and are critical for enabling frame-level semantic understanding.

To effectively supervise the multi-objective prediction heads of AHA, particularly the core *relevance head*, we construct a novel, large-scale dataset, named the **Human Intuition Highlight Dataset (HIHD)**. The construction of HIHD begins with the Mr.HiSum benchmark [15]: for each video entry therein, we retrieve its original full version from YouTube [34] via webscraping. Videos with fewer than 70,000 original views are subsequently discarded to ensure data quality. From the retained videos, we systematically sample frames at 1 fps to align with our model's visual processing rate. The corresponding YouTube replay counts (engagement scores [15]) are then normalized to a $[0, 1]$ range, serving as our primary relevance signal $r_t$, the ground truth scores for the relevance head's Smooth L1 loss (Eq. 1a). While this engagement based signal enables scalability far beyond manually labeled datasets, we acknowledge it is an imperfect proxy for true importance. It may introduce biases by amplifying content designed for high engagement (e.g., "clickbait") and misaligning with expert judgment in safety-critical domains. We provide detailed discussion of these limitations in Appendix I and Section 5, respectively. For our task-conditioned setting, relevant task objectives $Q$ are generated by programmatically transforming each video's original YouTube title into diverse natural language queries using predefined templates (e.g., a title "Exploring the Riemann Hypothesis" might become "What segment of the video addresses 'Exploring the Riemann Hypothesis'?"'); see Appendix H. Finally, to simulate real-world video stream degradation and enhance model robustness, we introduce "quality dropouts": 5–20% of each video's duration is randomly selected, and frames within these segments undergo perturbations such as resolution reduction, block noise, color banding, or blackouts, with corresponding dropout masks generated (detailed in Appendix E.1). Crucially, HIHD adopts the exact train/validation/test splits from Mr.HiSum to ensure fair comparability, and its training set explicitly excludes videos present in common highlight detection evaluation datasets.

The resulting **HIHD** comprises 22,463 videos, each with frame-level normalized engagement scores ($r_t$), a synthetic task objective $Q$, and quality dropout masks. This dataset provides rich, frame-level supervision specifically designed for training and evaluating task-conditioned OHD models like AHA. By combining large-scale implicit human engagement signals with synthetically generated task conditioning and targeted robustness augmentations, HIHD aims to foster the development of models that can model human intuition in dynamic, task-driven, and imperfect streaming environments.

To supervise the *informativeness head*, which predicts whether frame $f_t$ introduces new information, we adapt strategies from MMDuet [13], a streaming framework. Ground truth labels $i_t \in \{0, 1\}$ are derived from segment-level captions in the human-annotated subset of **Shot2Story** [35] and procedural videos from **COIN** [36]. For each segment, a "point of sufficient understanding" is randomly sampled between 50% and 75% of its duration. Frames from the 50% mark up to this point are labeled informative ($i_t = 1$); others before or after are labeled non-informative ($i_t = 0$). This reflects the intuition that early frames lack context and later ones become redundant once understanding is achieved. The informativeness head is trained using BCE loss (Eq. 3), with the hypothesis that this signal correlates with highlight moments in OHD. Crucially, our framework is designed to explicitly decouple this signal of informational novelty from task-relevance, using separate heads to learn these distinct concepts. We provide a detailed justification and a qualitative analysis demonstrating its effectiveness on a real-world robotics video in Appendix D.1.

To enhance semantic representations, we train an auxiliary *LM head* using the same **Shot2Story** and **COIN** annotations. At random timesteps, AHA generates a short caption for the current frame conditioned on prior context and the task prompt, supervised via next-token cross-entropy (Eq. 6). Unlike interactive systems (e.g., MMDuet), AHA does not re-inject generated text into the context or use it during inference. This preserves its unidirectional, non-dialogue streaming setup. The LM task solely improves the quality of hidden representations $h_t$ used by the highlight prediction heads (see Appendix D for details).

# 4 Experiments

This section details the comprehensive experimental evaluation of AHA. We first assess its core performance as an OHD model under strict streaming constraints on two standard HD benchmarks, TVSum and Mr.HiSum (Section 4.1). We then evaluate its robustness to common video degradations and conduct ablation studies to analyze the contributions of its key components (Section 4.2). To demonstrate its practical applicability in challenging real-world conditions, we further test AHA's capabilities on a long-form robotics video (Section 4.3), and generalization potential to other unoptimized video understanding tasks (Section 4.4). Our results are averaged over 5 runs.

## 4.1 Highlight Detection

The widely-used **TVSum** HD benchmark [14] provides multi-rater frame-level importance scores for 50 diverse videos. However, its small size can cause topic bias in standard splits, hindering reliable generalization assessment [15]. Therefore, to rigorously assess generalization from its pre-training (Sec. 3.3), we evaluate AHA on TVSum zero-shot (i.e., without TVSum-specific fine-tuning) and with a lightweight grid search. Following [10], we report Kendall's $\tau$ (ordinal association) and Spearman's $\rho$ (monotonic relationship) rank correlations. We also report top-5 mAP (mean Average Precision for top 5 summary segments) per established TVSum protocols.

On TVSum (Table 1), AHA establishes a new SOTA demonstrating remarkable performance even in a truly zero-shot setting. Using a fixed heuristic without any domain specific tuning, our model achieves 91.6 top-5 mAP, significantly outperforming the previous best tuned model, TR-DETR [19] (87.1 mAP). This zero-shot configuration also produces the most faithful overall frame ranking, setting a new SOTA on both Kendall's $\tau$ (0.304) and Spearman's $\rho$ (0.433).

Furthermore, performance on summary retrieval can be pushed even higher. By adapting the scoring parameters via a lightweight grid search on the TVSum validation set, the top-5 mAP is boosted to 93.0. This domain-adapted configuration slightly alters the global ranking but excels at the primary goal of identifying the most critical highlight segments.

Table 1: TVSum Performance. We report top-5 mAP, $\tau$, and $\rho$. 'Tuned?' indicates if fine-tuned on TVSum (Y) or not (N). Modalities: V (visual), T (text), A (audio). **Bold** is SOTA. (Per-category details: Appendix B.2).

| Model | Tuned? | Modality | mAP | Kendall $\tau$ | Spearman $\rho$ |
|---|---|---|---|---|---|
| Human [20] | N | V | – | 0.177 | 0.204 |
| PGL-SUM [18] | N | V | 57.1 | 0.206 | 0.157 |
| LLMVS [10] | N | V+T | – | 0.211 | 0.275 |
| UniVTG [17] | N | V | 84.6 | – | – |
| QD-DETR [37] | Y | V+A | 86.6 | – | – |
| TR-DETR [19] | Y | V+A | 87.1 | – | – |
| AHA (Zero-Shot) | N | V+T | 91.6 | **0.304** | **0.433** |
| **AHA (Domain-Adapted)** | N | V+T | **93.0** | 0.285 | 0.406 |

The large-scale **Mr.Hisum** HD benchmark [15] uses YouTube replay statistics ("most replayed" data reflecting broad viewer engagement) as scalable ground truth, forming a key component of our HIHD (Sec. 3.3). Since AHA's training (via HIHD) uses only Mr.Hisum's *training* split data, our evaluation on its *test set* is strictly on held-out data. Per protocol [15], we report mAP@50 and mAP@15 (top 50/15 ranked segments) to assess relevance assignment to frequently rewatched frames.

To specifically evaluate our relevance head on the task it was trained for, we use a scoring configuration that isolates its output ($\beta = 1$, with all other weights set to zero). On the Mr.Hisum test set (Table 2), this focused approach achieves a new SOTA of 64.19 mAP@50 and 32.66 mAP@15, a significant improvement (e.g., +8.3 mAP@50 over PGL-SUM [18]). These results validate that our relevance head, trained on large-scale engagement, successfully identifies salient moments correlated with user engagement under strict no future access constraints.

Table 2: Overall HiSum performance on the full test set. **Bold** highlights our SOTA results.

| Metric | SL-module [38] | iPTNet [39] | DSNet [40] | PGL-SUM [18] | AHA (Ours) |
|---|---|---|---|---|---|
| mAP@50 | 55.31 | 50.53 | 50.78 | 55.89 | **64.19** |
| mAP@15 | 24.95 | 22.74 | 24.35 | 27.45 | **32.66** |

## 4.2 Ablations

We conduct ablations on TVSum for its evaluation of streaming scoring and ranking; Mr.HiSum is omitted as it mainly tests the relevance head. AHA is tested with the optimal sliding window ($n = 2048$) unless otherwise specified. Core component and SinkCache configuration results are shown in Table 3. We also demonstrate the efficacy of AHA's video quality dropout training in Table 4 by evaluating its performance under various visual degradations.

**Head Importance.**    The decoupled prediction heads are crucial (Table 3, left). Removing the relevance ($\beta = 0$) or informativeness ($\alpha = 0$) heads severely degrades Top-5 mAP by 15.7 and 9.8 points, respectively, from our 93.0 mAP baseline. Omitting uncertainty ($\epsilon = 0$) results in a smaller drop (3.2 mAP points), suggesting it aids calibration but is less critical here than the other heads.

**Language Conditioning.**    Eliminating language conditioning (empty task string $Q$) significantly reduces Top-5 mAP by 11.8 points ($93.0 \rightarrow 81.2$) and drastically lowers rank correlations (e.g., S-$\rho$: $0.406 \rightarrow 0.342$). This highlights the critical role of persistent language grounding for task-conditioned HD in streaming video, where retaining the task objective enables AHA to maintain long-range semantic alignment. Furthermore, the model exhibits graceful degradation under imperfect conditioning. When tested with ambiguous (i.e., overly general) or entirely irrelevant prompts, performance declines proportionally rather than catastrophically, confirming that the language prompt strongly guides but does not dominate the underlying visual saliency detection (see Appendix B.5).

**Memory Mechanism Analysis.**    Our most significant architectural finding comes from ablating the memory mechanism itself (Table 3, right). We found that simpler strategies relying on only recent context ('Sliding Window Only') or only initial context ('Static Window Only') performed poorly. Our proposed **Dynamic SinkCache** outperforms not only these simpler methods but also an "Unbounded KV Cache" and the "Standard SinkCache", proving that a task-focused sink is the optimal memory strategy for this problem. The choice of a 2048 token window for recent context provides an excellent balance of performance and efficiency, as detailed in our window size analysis in Appendix F.2. The details of each memory mechanism can also be found in Appendix F.4

Table 3: Ablation study on TVSum. Left: Core component ablations. Right: Memory mechanism ablations. Our default model (**top row**) uses the Dynamic SinkCache.

| Variant | mAP | S-$\rho$ | K-$\tau$ |
|---|---|---|---|
| **AHA (Ours)** | **93.0** | **0.406** | **0.285** |
| $\alpha = 0$ | 83.2 | 0.341 | 0.237 |
| $\beta = 0$ | 77.3 | 0.321 | 0.221 |
| $\epsilon = 0$ | 89.8 | 0.401 | 0.278 |
| w/o $Q$ | 81.2 | 0.342 | 0.238 |

| Memory Mechanism | mAP | S-$\rho$ | K-$\tau$ |
|---|---|---|---|
| **Dynamic SinkCache (Ours)** | **93.0** | **0.406** | **0.285** |
| Standard SinkCache | 92.6 | 0.401 | 0.280 |
| Unbounded KV Cache | 91.7 | 0.400 | 0.277 |
| Sliding Window Only | 69.5 | 0.063 | 0.043 |
| Static Window Only | 63.2 | 0 | $-0.01$ |

**Impact of Video Quality Dropout Training.** A key design goal for AHA is reliable performance despite visual degradations common in real-world streaming. Its training incorporates video quality dropout mechanisms (Appendix E.1) specifically to build resilience against such artifacts. To demonstrate the efficacy of this training approach, we evaluated AHA on the TVSum dataset under both clean conditions and with several simulated visual degradations [41]: *color banding*, *block noise*, *quality degradation*, and *blackout*, each applied to 20% of frames within each video. As detailed in Table 4, AHA exhibits notable resilience. For instance, its Top-5 mAP drops by only ($\Delta_{0.4}$) and ($\Delta_{1.8}$) percentage points when subjected to *color banding* and *block noise*, respectively. Even when faced with more severe artifacts like *quality* degradation and complete *blackout*, AHA maintains strong absolute performance with mAP scores of 88.9 ($\Delta_{4.1}$) and 88.2 ($\Delta_{4.8}$). These findings confirm that the video quality dropout strategy employed during AHA's training is effective in preparing it for

imperfect visual inputs. This enables graceful degradation and underscores its potential for reliable deployment in real-world streaming environments where video quality can be unpredictable.

Table 4: Robustness to video corruptions on TVSum (Top-5 mAP). $\Delta$ indicates drop from clean.

|  | Clean | +ColorBanding | +BlockNoise | +Quality | +Blackout |
|---|---|---|---|---|---|
| **AHA (Ours)** | **93.0** | **92.6** ($\Delta_{0.4}$) | **91.2** ($\Delta_{1.8}$) | **88.9** ($\Delta_{4.1}$) | **88.2** ($\Delta_{4.8}$) |

## 4.3 Real-World Evaluation on Long-Form Robotics Video

We evaluate AHA on video from the SCOUT dataset [16], a long-horizon (20+ min), egocentric video captured during indoor robot navigation trials from human-robot collaborative exploration exercises.[3] Unlike web videos, SCOUT features continuous footage with no cuts, degraded quality (e.g., static, warping), and sparse, mission-relevant events, providing a challenging, real-world testbed for OHD.

AHA generates highlight scores ($\hat{y}_t$) in real-time. To facilitate qualitative analysis and enable the creation of a structured highlight reel from these continuous online scores, we apply Savitzky-Golay smoothing [42] followed by peak detection as a *post-processing* step to isolate segments of high salience. Ground truth video annotations were established by domain experts by aligning these predicted peaks with human-issued navigation commands (obtained from experiment transcripts) and key visual transitions observed in the video footage. In an 8-minute analysis (Fig. 2), 16 of 18 predicted peaks aligned with human-issued commands or meaningful actions (e.g., "robot take a better picture of the shoes", "enter the room"). Despite **the aforementioned noise and visual corruption, AHA remained stable.** This stability is consistent with its designed resilience to such artifacts (as demonstrated in Sec. 4.2 and Table 4), and AHA also showed strong alignment with semantic shifts. These results, while preliminary in application to this dataset, suggest that AHA can detect high-salience moments in real-time, supporting both operator alerting and automated highlight generation in field robotics deployments (see Appendix G for more details).

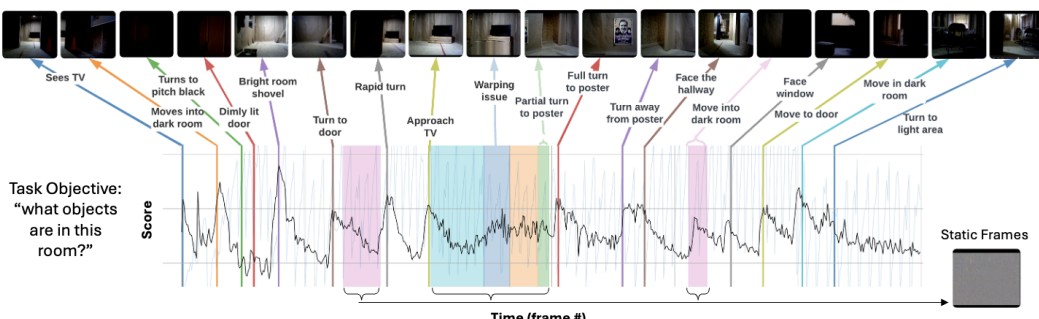

Figure 2: SCOUT results. Colored lines mark annotated events from video (e.g., room entry, turns). Highlighted regions indicate degraded video. Black line is AHA's predicted highlight scores.

## 4.4 Generalization to Broader Streaming Video Understanding

Beyond its primary application in highlight detection, AHA's architectural design and learned representations demonstrate strong potential for broader video understanding. When evaluated on a streaming moment retrieval (MR) protocol [13] using the Charades-STA dataset [43], AHA achieves SOTA performance among streaming methods, yielding 50.7% R@1 at an IoU of 0.5, and 27.9% R@1 at an IoU of 0.7. This result highlights the robustness of our approach for fine-grained temporal understanding in streaming video. Full details of this streaming MR evaluation, including comparative results, are provided in Appendix B.1. Additionally, AHA's capabilities on other video-language tasks such as dense captioning and multi-answer grounding are discussed in Appendix B.

---

[3]A subset of video frames available in SCOUT repository. Full videos planned for near-term public release.

# 5 Conclusion

We introduced AHA, a real-time, task-conditioned OHD framework. Its lightweight prediction heads and novel Dynamic SinkCache-based memory achieve SOTA performance on standard HD benchmarks, remarkably outperforming even traditional offline methods while maintaining constant computational cost across arbitrarily long video streams. Trained on our HIHD data, derived from user engagement signals and task-conditioned prompts, AHA aligns with human-like intuition for relevance. We validated its robustness on standard benchmarks and challenging real-world settings, including streaming MR and the long-horizon SCOUT robotics dataset, demonstrating AHA's capability for consistent, task-relevant understanding in noisy, real-world conditions where offline processing is often infeasible.

Looking ahead, AHA offers a scalable solution for intelligent agents requiring real-time, context-aware video understanding, such as for surveillance drones, satellite analysis, embedded systems, and disaster response. Our ongoing work is developing further analysis of the SCOUT videos, and extending AHA to publicly available drone footage from disaster response efforts (e.g., wildfire monitoring), where its online, task-conditioned highlight detection can be tailored to aid responders and investigators in identifying mission-critical information from continuous video streams.

**Limitations and Future Work.** Although AHA achieves strong results, we identify opportunities for future improvement.

*Uncertainty Modeling:* Our uncertainty head is trained without ground-truth uncertainty labels due to the subjective nature of highlights and the difficulty of capturing annotator confidence at scale, limiting interpretability in high-stakes settings. Future work could explore supervised, contrastive, or calibrated approaches. For instance, datasets with human confidence scores, such as MultiVENT-G [44], offer a promising path towards direct supervision (see Appendix I). Despite this, our model demonstrates improved performance when incorporating uncertainty into its scoring.

*Training Efficiency and Backbone Generalization:* High compute costs restricted our architectural ablations, including the validation of the AHA framework on a wider range of vision-language backbones (see Appendix C.5 for our selection rationale). Future work could explore distilled variants of AHA to improve runtime efficiency, which would facilitate this broader testing and confirm the framework's adaptability across different underlying models.

*Static Inference Weighting:* Our framework relies on a static weighting scheme for inference, a design choice empirically validated in our ablations (Appendix C.4), where it proved more effective than the dynamic alternatives we tested. While this modular approach yields SOTA performance, the exploration of more sophisticated adaptive weighting mechanisms remains a compelling direction for future research. Expert annotated datasets like MultiVENT-G [44] could provide an important testbed for validating any weighting strategy against human defined importance (see Appendix I).

*Memory Constraints in Training:* Training uses fixed-length windows without persistent memory across segments for efficient batching. While the Dynamic SinkCache at inference provides stable, bounded memory, future work could explore augmenting training with recurrent or retrieval-based memory mechanisms to potentially enhance global reasoning capabilities learned by the model.

**Ethical Considerations and Broader Impact.** Despite its benefits for applications like disaster response, AHA could be misused in surveillance contexts or amplify societal biases if trained on biased data. We recommend its deployment with privacy-preserving measures (e.g., blur filters for faces), robust access controls, and domain-specific ethical audits. To guide responsible research and application, a code of conduct will accompany our public repository. As this technology develops, we encourage continued open discussion regarding its ethical deployment, particularly in sensitive domains such as public safety, surveillance, or defense.

## Acknowledgments and Disclosure of Funding

This research was conducted while the first author was a research fellow at the DEVCOM Army Research Laboratory (ARL), supported by an Army Educational Outreach Program (AEOP) fellowship administered through the Rochester Institute of Technology. We thank ARL for additional compute support and mentorship, and the anonymous reviewers for their valuable feedback, which improved the paper, including the development of the Dynamic SinkCache.

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

# A  Training Hyperparameters

Table 5 gives the full set of hyperparameters used to fine-tune AHA on the Qwen-7B backbone.

Table 5: Key hyperparameters for training AHA.

| Category | Hyperparameter (Value) |
|---|---|
| *Optimization* | |
| Optimizer | AdamW [45] |
| Betas (optimizer) | (0.9, 0.999) |
| Epsilon (optimizer) | $1 \times 10^{-8}$ |
| Weight decay | 0.0 |
| Learning rate | $2 \times 10^{-5}$ |
| LR scheduler | Cosine decay with linear warmup |
| Warmup ratio | 0.05 (0 warmup steps) |
| Gradient norm clipping | 1.0 |
| Gradient checkpointing | Enabled |
| | |
| *Batching* | |
| Per-device train batch size | 1 |
| Gradient accumulation steps | 2 (effective batch size = 2) |
| Num epochs | 1 |
| | |
| *Precision & Acceleration* | |
| BF16 training | Enabled |
| DeepSpeed | zero2 [46] + CPU offload |
| Attn implementation | Flash Attention2 [47] |
| | |
| *Data loading* | |
| Dataloader workers | 4 |
| Pin memory | True |
| Drop last batch | False |
| | |
| *Video preprocessing* | |
| Frame rate | 1 fps |
| Frame resolution | $384 \times 384$ |
| Pooling stride | 4 |
| Frame tokens (#) | 49 |
| Token pooling dims | [7, 7] |
| | |
| *Model backbones* | |
| LLM backbone | `lmms-lab/llava-onevision-qwen2-7b-ov` |
| Vision backbone | `google/siglip-large-patch16-384` |
| Multimodal projector | 3×3 conv + linear layers |
| | |
| *Losses & regularization* | |
| Stream loss weight | 1.0 |
| TV loss window | 49 |
| | |
| *Saving & logging* | |
| Save strategy | steps (every 25 steps) |
| Save total limit | 5 checkpoints |
| Logging strategy | steps (every 1 step) |

## A.1  Implementation Details

We fine-tune AHA using Low-Rank Adaptation (LoRA) [48] on a frozen Qwen-2.7B backbone for one epoch. Training was performed on 3 compute nodes, each with 2×NVIDIA A6000 GPUs (48GB VRAM), totaling 6 GPUs. The full training run took approximately 28 hours. Videos were sampled at 1 fps for both training and inference. AHA was trained using PyTorch 2.5.1, Transformers 4.49.0, and CUDA 12.4 on Ubuntu 22.04. Training runs were executed on Paperspace, with all checkpoints and video data stored in an Amazon S3 bucket.

- **LLM backbone:** Frozen Qwen-2.7B [29] (`lmms-lab/llava-onevision-qwen2-7b-ov`).
- **Vision encoder:** Frozen SigLIP Large [27] (`google/siglip-large-patch16-384`).

- **MM projector:** Single linear layer `nn.Linear(mm_hidden_size, hidden_size)` mapping each 1152-d patch to Qwen's 3584-d hidden space.

## A.2 Inference Performance

We conducted a detailed performance analysis of our framework on a 1062 second video (~17 minutes) using two NVIDIA A6000 GPUs. The system achieved a sustained throughput of 1 frame per second (FPS), demonstrating high efficiency with 100% peak GPU utilization and 90% peak memory controller utilization. During this process, the framework consumed a peak of 30.49 GB of VRAM across both GPUs and operated well within safe thermal limits at a peak temperature of 65°C, all while maintaining a minimal system RAM footprint of 3.66 GB. While this establishes a strong performance baseline, the 1 FPS rate means that in a live scenario, the system would "drift" and fall behind the incoming video feed. Therefore, for real-time deployment, implementing logic to strategically skip frames would be necessary to keep the analysis on track. Given that our framework already leverages the available compute effectively, it is a strong candidate for such optimizations to achieve the higher throughput needed for live applications.

# B  Additional Results

## B.1  Streaming Moment Retrieval

We follow the streaming moment retrieval (MR) protocol introduced in MMDuet [13], applying AHA to Charades-STA [43] and treating frame-level relevance scores as soft temporal indicators. Using a smoothing window $w$ as in prior work, we compute R@1 at IoU thresholds of 0.5 and 0.7.

As shown in Table 6, AHA with $w = 8$ achieves the highest temporal grounding performance on Charades-STA, attaining 50.7% R@0.5 and 27.9% R@0.7. This constitutes an absolute improvement of 8.3 and 9.9 points, respectively, over the strongest baseline (MMDuet with $w = 8$), highlighting the benefits of our direct frame-level scoring and streaming-oriented design, even in the absence of span-level supervision.

To further contextualize these results, we train two additional streaming MR baselines following MMDuet's framework [13], using the same initialization (LLaVA-OneVision [49]), training data (MMDuetI [13]), and learning schedules. We reformat the data into the interaction and segment representation formats used by TimeChat [50] and VTimeLLM [51], yielding LLaVA-OV-TC and LLaVA-OV-VT. Comparisons to these variants further validate the advantages of our frame-level design.

We note, however, that this formulation is still an approximation of full moment retrieval. Accurate span localization under strict streaming constraints remains an open challenge. Extending AHA with autoregressive span prediction or memory aware temporal boundary modeling is a promising direction for future work.

Table 6: Performance on Charades-STA for temporal grounding.

| Metric | VTG-LLM [52] | LLaVA (OV-TC) | LLaVA (OV-VT) | MMDuet [13] | MMDuet ($w = 8$) | AHA (Ours) | AHA ($w = 8$) |
|---|---|---|---|---|---|---|---|
| R@0.5 | 33.8 | 33.1 | 36.5 | 27.3 | 42.4 | 42.8 | **50.7** |
| R@0.7 | 15.7 | 12.4 | 12.3 | 2.1 | 18.0 | 18.1 | **27.9** |

## B.2  TVSum's Categorical Evaluation

In addition to our overall Top-5 mAP results, we analyze performance across TVSum's ten activity categories [14]: Changing Vehicle Tire (VT), Getting Vehicle Unstuck (VU), Grooming an Animal (GA), Making Sandwich (MS), Parkour (PK), Parade (PR), Flash Mob Gathering (FM), Bee Keeping (BK), Attempting Bike Tricks (BT), and Dog Show (DS). As shown in Table 7, our full multimodal model (V+T) achieves a new state-of-the-art in nearly every category, with particularly large gains in visually complex tasks like Changing Vehicle Tire.

Table 7: Top-5 mAP (%) on TVSum categories. **Bold** indicates state-of-the-art per category.

| Method | Modality | VT | VU | GA | MS | PK | PR | FM | BK | BT | DS | Avg |
|---|---|---|---|---|---|---|---|---|---|---|---|---|
| QD-DETR [37] (V) | V | 88.2 | 87.4 | 85.6 | 85.0 | 85.8 | 86.9 | 76.4 | 91.3 | 89.2 | 73.7 | 85.0 |
| UniVTG [17] | V | 83.9 | 85.1 | 89.0 | 80.1 | 84.6 | 87.0 | 70.9 | 91.7 | 73.5 | 69.3 | 81.0 |
| TR-DETR [19] (V) | V | 89.3 | 93.0 | 94.3 | 85.1 | **88.0** | 88.6 | 80.4 | 91.3 | 89.5 | 81.6 | 88.1 |
| QD-DETR (V+A) | V+A | 87.6 | 91.7 | 90.2 | **88.3** | 84.1 | 88.3 | 78.7 | 91.2 | 87.8 | 77.7 | 86.6 |
| TR-DETR (V+A) | V+A | 90.6 | 92.4 | 91.7 | 81.3 | 86.9 | 85.5 | 79.8 | 93.4 | 88.3 | 81.0 | 87.1 |
| AHA (Ours) | V+T | **98.3** | **99.2** | **99.4** | 84.8 | 81.2 | **94.8** | **97.4** | **94.2** | **93.1** | **87.6** | **93.0** |

## B.3 Multi-Answer Grounded Video Question Answering

The Multi-Answer Grounded Video Question Answering (MAGQA) benchmark [13] extends conventional Video QA by requiring models to generate multiple answers at semantically relevant time points within a single video, rather than a single response per question. In MAGQA, each question corresponds to $n_{turns}$ ground-truth answer turns, each defined by a start time $start_q$, an end time $n_{turns}$, and an answer text $gold_q$. Models must decide, at each frame, whether to respond based on the sum of informative and relevance scores exceeding a threshold $t$, and then produce the answer in real-time. Performance is measured using the *in-span score*, which combines textual relevance (scored 1–5 via an LLM) with temporal accuracy by averaging the scores of all predicted answers falling within each ground-truth interval and then averaging across intervals. This setup simulates realistic streaming video comprehension, emphasizing both promptness and answer correctness without access to future frames.

Our model attains an in-span score of 2.42 (GPT-scored) at $t = 0.5$ and 2.37 at $t = 0.3$, compared to MMDuet's peak of 2.93 (Table 8), indicating that AHA can still produce timely, relevant multi-answer responses even without task-specific training. After deduplication, we average only about 2.02–2.09 unique turns per video, despite generating over 30 raw turns, showing that our streaming design reliably spots answerable moments but tends to repeat predictions when it isn't explicitly optimized for MAGQA. This highlights both the versatility of our framework in auxiliary QA tasks and the opportunity to further improve answer diversity and precision through dedicated fine-tuning.

Table 8: MAGQA evaluation results: In-span score and response turns

| Model | In-Span Score (LLaMA / GPT) | # Turns (w/o. / w/. dedup) |
|---|---|---|
| LLaVA-OV-TC | 2.92 / 2.79 | 3.4/1.9 |
| LLaVA-OV-VT | 2.94 / 2.78 | 5.4/2.2 |
| MMDuet [13] | | |
| w/ $t = 0.6$ | 2.46 / 2.33 | 13.7/4.0 |
| w/ $t = 0.5$ | 2.77 / 2.61 | 18.4/5.3 |
| w/ $t = 0.4$ | 3.00 / 2.81 | 23.0/6.6 |
| w/ $t = 0.3$ | **3.13 / 2.93** | 27.0/7.6 |
| AHA (Ours) | | |
| w/ $t = 0.5$ | 2.68 / 2.42 | 30.55 / 2.02 |
| w/ $t = 0.3$ | 2.63 / 2.37 | 34.19 / 2.09 |

## B.4 Dense Video Captioning

We evaluate on the YouCook2 dense video captioning benchmark [53], where models must detect and describe $\sim 8$ procedural steps in minute-long cooking videos by outputting, for each step, a start time, end time, and caption. Following MMDuet [13], we accumulate a per-frame "need response" score (the sum of informative and relevance heads) and emit a caption whenever this sum exceeds a threshold $s$ (we set $s = 2$). Since frames themselves do not explicitly mark step boundaries, we heuristically assign the previous and current response times as the start and end of each segment, and merge adjacent steps with identical captions.

Table 9 compares performance. Even without any DVC-specific fine-tuning, AHA produces competitive captions in real-time, achieving an F1 of 15.1%, demonstrating its versatility across auxiliary tasks. However, unlike MMDuet's "rm. prev. resp." trick, which significantly reduces redundancy, our streaming design still tends to repeat captions, reflecting the need for dedicated training or more sophisticated boundary modeling to fully match specialized DVC pipelines.

Table 9: Performance on YouCook2 dense video captioning.

| Method | SODA$_c$ | CIDEr | F1 |
|---|---|---|---|
| TimeChat [50] | 1.2 | 3.4 | 12.6 |
| VTG-LLM [52] | 1.5 | 5.0 | 17.5 |
| LLaVA-OV-TC | 1.9 | 3.3 | 21.8 |
| LLaVA-OV-VT | 2.5 | 6.7 | 14.0 |
| MMDuet [13] | 2.4 | 5.7 | 19.2 |
| + rm. prev. resp. | **2.9** | **8.8** | **21.7** |
| **AHA (Ours)** | 1.4 | 3.2 | 15.1 |

### B.5 Robustness to Imperfect Task Conditioning

To assess the framework's robustness to variations in task conditioning, we conducted a quantitative analysis on the TVSum dataset using ambiguous and irrelevant prompts. An **ambiguous prompt** was defined as a high-level categorical description of the specific task (e.g., using "Vehicle Maintenance" for a video on changing tires). An **irrelevant prompt** was defined as a task description sampled from a video in a completely different category.

Performance was measured by the change in top-5 mAP relative to the baseline score achieved with the original, specific prompt (93.0 mAP). The results, summarized in Table 10, demonstrate graceful degradation. With an ambiguous prompt, performance decreased by only 1.1 mAP points, indicating the model can generalize to broader task descriptions. When given an entirely irrelevant prompt, performance dropped by a more significant, yet not catastrophic, 9.7 mAP points. This confirms that while the model is strongly guided by the task objective, its learned visual representations retain a strong sense of inherent saliency.

Table 10: Impact of prompt quality on TVSum performance (Top-5 mAP).

| Prompt Type | Top-5 mAP | Change ($\Delta$) |
|---|---|---|
| Standard (Specific) | 93.0 | Baseline |
| Ambiguous | 91.9 | -1.1 |
| Irrelevant | 83.3 | -9.7 |

## C   Supplementary Methodological Details

This section provides additional details and justifications for certain design choices described in the main paper's Methodology section (Section 3), which were condensed for brevity due to page limits.

### C.1   Training Objectives: Head Details and Justifications

**Relevance Head - TV Loss Motivation.**   The motivation for incorporating the total variation (TV) loss (Eq. 1b in the main paper) stems from observing the structure of human engagement signals often used for highlight supervision, such as aggregated user replay statistics (see Figure 3). These signals frequently exhibit smooth, bell-shaped distributions centered on replayed segments. The TV loss encourages our relevance predictions $\hat{r}_t$ to match these smooth trends characteristic of engagement, complementing the point-wise Smooth L1 loss [30] (Eq. 1a) while aiming to avoid over-smoothing across genuine sharp transitions in content relevance. The term $v_t$ acts as a binary mask ensuring the penalty applies only to adjacent valid predictions.

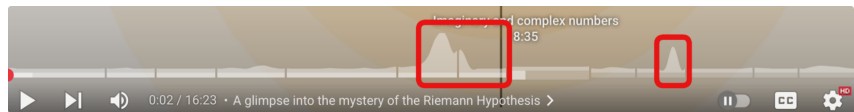

Figure 3: YouTube replay distribution, adapted from the Mr.Hisum [15]. Peaks in replay volume (vertical axis), forming "bell curves," indicate frequently rewatched segments. These high-engagement areas serve as the primary supervision signal for training our relevance prediction head.

**Informativeness Head - Rationale and Repurposing.** While prior work in dialog-based VideoLLMs [13, 12] often utilizes informativeness scores to trigger language generation or manage conversational turns, we adapt this underlying intuition as a direct learning signal specifically for the highlight detection (HD) task. By supervising the model (Eq. 3) to explicitly recognize temporally novel versus redundant frames, we encourage the development of stronger temporal reasoning capabilities, which is beneficial for accurate highlight estimation over extended periods.

**Uncertainty Head - Rationale and Potential Applications.** The introduction of the uncertainty head is crucial for addressing the challenges of the online, streaming setting. Since the model must predict relevance $\hat{r}_t$ at time $t$ based only on past and current information (partial observability), its ability to judge the long-term significance of a frame is inherently limited. Training the model to predict its own uncertainty via log variance of the relevance score, using the negative log-likelihood objective in Eq. 4, explicitly models this limitation. Specifically, the model outputs a raw log-variance $\hat{l}_t = W_u h_t$, which is clamped for numerical stability and then exponentiated to obtain the predicted variance $\hat{\sigma}_t^2$. During inference, we use the clamped log-variance $\hat{l}_{t,c}$ as the uncertainty score $\hat{u}_t$, as it is more stable and interpretable for downstream use. As noted in the main text, this is, to our knowledge, the first application of such probabilistic uncertainty modeling in OHD.

Beyond the immediate model training, the resulting uncertainty scores $\hat{u}_t$ can potentially support downstream applications such as adaptive decision thresholds, mechanisms for deferring judgment on low-confidence frames, or reliability-aware resource allocation when processing multiple video streams.

For a detailed justification of this architecture, including comparisons to alternative uncertainty estimation techniques such as Monte Carlo Dropout and Bayesian inference, see Appendix C.3.

**LM Head - Design Choice Justification.** The auxiliary LM head (Eq. 6) aims to foster semantically rich hidden representations. In contrast to conversational models that might inject generated text back into the context [12, 13], we deliberately avoid this feedback loop. Our focus is on efficient, unidirectional frame-wise scoring for HD, not multi-turn interaction. This decoupling enhances efficiency and avoids reliance on implicit conversational structures that may not align well with continuous, non-interactive video streams. The LM task serves solely to improve multimodal alignment in the representations used by the primary scoring heads.

### C.2 Loss Function Weights

The total loss function used for training AHA is a weighted sum of the objectives from the different prediction heads, as defined in Eq. 7:

$$\mathcal{L}_{\text{total}} = \lambda_{\text{r-total}}\mathcal{L}_{\text{relevance-total}} + \lambda_{\text{i}}\mathcal{L}_{\text{info}} + \lambda_{\text{u}}\mathcal{L}_{\text{uncertainty}} + \lambda_{\text{LM}}\mathcal{L}_{\text{LM}}$$

where $\mathcal{L}_{\text{relevance-total}}$ itself combines the base relevance loss and the total variation loss: $\mathcal{L}_{\text{relevance-total}} = \mathcal{L}_{\text{relevance}} + \lambda_{\text{TV}}\mathcal{L}_{\text{TV}}$ (Eq. 2).

The weights ($\lambda$) were determined based on the relative importance of each task, considerations of class imbalance, the role of auxiliary objectives, and preliminary experiments. The final fixed weights used throughout training are detailed below, following a general strategy of up-weighting critical or difficult tasks and down-weighting auxiliary or regularizing terms:

- **Relevance Loss Weight ($\lambda_{\text{r-total}} = 8.0$):** The total relevance loss ($\mathcal{L}_{\text{relevance-total}}$), which includes the primary SmoothL1 regression objective (Eq. 1a), is assigned the highest weight (8.0). This emphasizes the main goal of the model: accurately predicting task-conditioned

highlight relevance. This aligns with multi-task learning principles where primary task losses are often weighted higher [54].

- **Internal TV Loss Weight ($\lambda_{\mathbf{TV}} = 0.05$):** Within the $\mathcal{L}_{\text{relevance-total}}$ term (Eq. 2), the total variation loss component (Eq. 1b) is weighted relatively low ($\lambda_{\text{TV}} = 0.05$). This ensures it functions as a regularizer, encouraging temporal smoothness in predictions without dominating the main regression signal from $\mathcal{L}_{\text{relevance}}$.

- **Informativeness Loss Weight ($\lambda_{\mathbf{i}} = 0.5$):** The informativeness head's BCE loss (Eq. 3) is assigned a significant weight (0.5). This decision addresses the substantial class imbalance inherent in many HD tasks, where non-informative frames often form the vast majority. By up-weighting this loss, we ensure the model remains sensitive to detecting rarer informative frames, drawing inspiration from methods like Focal Loss [55] that effectively give more weight to harder examples or minority classes.

- **Uncertainty Loss Weight ($\lambda_{\mathbf{u}} = 0.1$):** The uncertainty head's NLL loss (Eq. 4) is considered an auxiliary objective. Its main purpose during training is to learn to predict the variance ($\sigma_t^2$) associated with the relevance prediction, rather than directly driving the relevance value itself. Consequently, it receives a small fixed weight (0.1), reflecting its supporting role relative to the primary relevance task (approximately 80 times smaller weight).

- **Language Modeling Loss Weight ($\lambda_{\mathbf{LM}} = 0.2$):** The LM head's cross-entropy loss (Eq. 6) also serves an auxiliary function, primarily aimed at enriching the model's internal multimodal representations, analogous to how models like BLIP [56] benefit from combined vision-language objectives during pre-training. Unlike models where text generation might be a primary output (e.g., [12]), here it supports the main scoring task and is weighted accordingly (0.2).

- **Variance Diversity Weight ($\lambda_{\mathbf{div}} = -e^{-3}$):** Small constant regularizing the uncertainty loss (Eq. 5a).

This multi-objective setup, common in complex vision-language tasks, allows the model to learn diverse but complementary skills necessary for effective highlight detection. The chosen weights reflect a balance aimed at prioritizing the core relevance prediction while leveraging the benefits of auxiliary signals for robustness, temporal understanding, and uncertainty awareness.

## C.3 Uncertainty Modeling Design

**Motivation.** In Online Highlight Detection (OHD), the model observes a video frame-by-frame and must immediately judge whether a frame is task-relevant, without seeing the future. This partial observability inherently limits predictive certainty. For example, the current frame may only gain meaning retroactively (e.g., as a prelude to an event). To address this, we introduce a lightweight uncertainty head that models *aleatoric uncertainty* (input-dependent uncertainty), i.e., the ambiguity in predictions stemming from incomplete observations. The head outputs a log-variance value $\hat{l}_t = W_u h_t$ at each timestep, predicting the uncertainty of the corresponding relevance score $\hat{r}_t$.

**Architecture and Loss.** We adopt a standard heteroscedastic regression formulation [57], treating the ground-truth relevance $r_t$ as sampled from a Gaussian distribution with mean $\hat{r}_t$ (the relevance head output) and predicted variance $\hat{\sigma}_t^2 = \exp(\hat{l}_{t,c})$. Here, $\hat{l}_{t,c}$ is the clamped log-variance for numerical stability. The primary training objective for the uncertainty head is the Gaussian negative log-likelihood (NLL) [25]:

$$\mathcal{L}_{\text{NLL}} = \frac{(r_t - \hat{r}_t)^2}{2\hat{\sigma}_t^2 + \delta} + \frac{1}{2}\log(2\pi\hat{\sigma}_t^2 + \delta)$$

We follow best practices from prior work [58] by predicting log-variance rather than variance directly, ensuring positivity and improving numerical stability.

**Preventing Mode Collapse.** A well-known issue with heteroscedastic models is the risk of degenerate solutions where the network minimizes the NLL loss by predicting arbitrarily high variances, thereby flattening the likelihood [32]. To mitigate this, we introduce a regularization term encouraging diversity in predicted uncertainties:

$$\mathcal{L}_{\text{div}} = -\lambda_{\text{div}} \cdot \text{std}(\{\hat{l}_{i,c}\}_{i \in \text{batch}})$$

The final uncertainty loss is defined as:

$$\mathcal{L}_{\text{uncertainty}} = \max(0, \mathbb{E}[\mathcal{L}_{\text{NLL}}] + \mathcal{L}_{\text{div}})$$

This discourages the model from assigning identical uncertainty across all frames and promotes calibration across predictable and ambiguous scenes.

**Comparisons to Alternative Approaches.**

- **Monte Carlo Dropout (MC Dropout)** [59]: Applies dropout during inference to simulate an ensemble. Multiple stochastic forward passes yield a distribution over predictions. While simple and widely used, MC Dropout primarily captures epistemic (model) uncertainty and requires multiple passes per frame, a poor fit for real-time streaming.
- **Bayesian Neural Networks (BNNs)** [60]: Learn distributions over weights via variational inference. While theoretically appealing, BNNs incur high computational cost and complex training [61]. Their benefit is mostly in epistemic uncertainty, which is less central than aleatoric uncertainty in the OHD setting.
- **Deep Ensembles** [61]: Combine predictions from independently trained models. This method produces state-of-the-art uncertainty estimates but is expensive at inference, requiring $M$ forward passes (where $M$ is the number of NNs in the ensemble). Ensembles are known to produce well-calibrated results but are impractical for streaming environments.

**Why Log-Variance Prediction?** Compared to these alternatives, our design:

- Requires **only a single forward pass**, making it suitable for high-frequency, low-latency inference.
- Provides **per-frame aleatoric uncertainty**, allowing the model to express ambiguity due to missing future context.
- Outputs **interpretable uncertainty scores** (clamped log-variance) that are usable downstream for decision deferral or confidence-weighted policies.
- Avoids trivial high-variance collapse via a **diversity promoting regularizer**.

This decision is further supported by recent work on heteroscedastic modeling in noisy classification settings [62], which shows improved robustness and calibration when modeling log-variance directly.

**Summary.** We favor log-variance prediction with NLL loss due to its interpretability, ability to model aleatoric uncertainty under partial observability, and compatibility with efficient online inference. Alternative approaches incur significant overhead or focus on epistemic uncertainty, which is secondary in our setting. Our approach allows AHA to not only predict whether a frame is relevant, but also how confident it is in that decision, a vital feature for intelligent agents in real-world deployments.

## C.4 Highlight Score Fusion

This subsection details the formulation of our highlight scoring function, the theoretical principles motivating its design, and the empirical validation for our choice of weighting scheme.

### C.4.1 Scoring Function Formulation

To compute the final scalar highlight score $\hat{y}_t$ per frame $f_t$, we fuse the outputs of the relevance ($\hat{r}_t$), informativeness ($\hat{i}_t$), and uncertainty ($\hat{u}_t$) heads. We adopt a piecewise linear, uncertainty-aware scoring rule that penalizes predictions made with high uncertainty:

$$\hat{y}_t = \begin{cases} \alpha\hat{i}_t + \beta\hat{r}_t, & \text{if } \hat{u}_t \leq \tau_u \quad \text{(low uncertainty)} \\ \alpha\hat{i}_t + \beta\hat{r}_t - \epsilon(\hat{u}_t - \tau_u), & \text{if } \hat{u}_t > \tau_u \quad \text{(high uncertainty)} \end{cases} \qquad (9)$$

Here, $\alpha$ and $\beta$ weight the informativeness and relevance signals, $\tau_u$ is an uncertainty threshold, and $\epsilon$ controls the penalty for predictions exceeding this threshold.

### C.4.2 Theoretical Motivation

This scoring function is designed for the specific challenges of Online Highlight Detection (OHD), where the model operates under partial observability. The core motivation mirrors principles from selective prediction and risk-aware decision-making [61, 63]: a system should only act (e.g., flag a highlight) when it is confident. The uncertainty signal $\hat{u}_t$ serves as a confidence gate. Below the threshold $\tau_u$, scores are computed normally; above it, a linear penalty is applied to down-weight uncertain decisions, implementing a simple risk-averse policy.

The piecewise linear design is deliberately chosen because it is:

1. **Modular:** Each head is trained independently, enabling post-hoc fusion without complex joint optimization.
2. **Interpretable:** Highlight scores are directly influenced by human-readable weights and a confidence gate.
3. **Stable:** The threshold provides consistent behavior in streaming conditions, avoiding erratic outputs from minor uncertainty fluctuations.
4. **Efficient:** It requires minimal computation per frame, making it ideal for real-time inference.

### C.4.3 Justification of Static Weighting

To justify our choice of a static weighting scheme, we compared it against more complex, dynamic alternatives on the TVSum benchmark. We evaluated three primary strategies, representing a spectrum from robustness to domain-specific optimality: (1) two unstable **dynamic methods**, (2) a robust **static zero-shot heuristic**, and (3) our top-performing, domain-adapted **static grid search**.

Table 11: Comparison of scoring mechanisms on TVSum (Top-5 mAP).

| Method | Top-5 mAP | Notes |
|---|---|---|
| Dynamic (MLP Gating) | 87.9 | Unstable, high variance |
| Dynamic (EMA Adaptor) | 87.5 | Unstable, high variance |
| Static (Zero-Shot Heuristic) | 91.6 | Most robust, SOTA baseline |
| **Static (Grid Search)** | **93.0** | **Data-sensitive, optimal performance** |

The results in Table 11 empirically validate the two configurations presented in our main results. The dynamic methods proved unstable and underperformed. In contrast, the **Static (Zero-Shot Heuristic)**, which uses the fixed parameters $\alpha = 0.7, \beta = 1.0, \epsilon = -2.9$, and $\tau_u = 0.3$, provides a highly robust baseline that already surpasses prior state-of-the-art. The **Static (Grid Search)** method further boosts performance to achieve the optimal score, confirming the value of lightweight domain adaptation, though its outcome is sensitive to the validation data.

The specific domain-adapted parameters found via this grid search, which were used to achieve our highest reported results, are detailed in Table 12.

Table 12: Optimal hyperparameters per dataset. Note: to reproduce these results you will likely need to run your own grid search.

| Dataset | $\alpha$ | $\beta$ | $\epsilon$ | $\tau$ |
|---|---|---|---|---|
| TVSum [14] | 0.667 | 1.357 | 3.571 | 0.077 |
| Mr.HiSum [15] | 0.000 | 1.778 | 0.714 | 0.040 |
| Charades [43] | 0.888 | 2.0 | -2.143 | 0.040 |
| SCOUT [16] | 0.200 | 1.556 | 1.000 | 0.053 |

### C.4.4 Comparison to Alternative Fusion Approaches

Our simple, modular scoring function was chosen over other common fusion techniques for its suitability in a streaming context.

- **Learned Fusion:** Using a neural network to learn the fusion function sacrifices the interpretability and modularity that are critical for domain adaptation and risks overfitting.
- **Attention-Based Weighting:** A dynamic attention mechanism over the heads introduces additional parameters and potential instability in a streaming setting, complicating calibration.
- **Confidence-Weighted Blending:** Using a continuous function (e.g., sigmoid scaling) is more complex to tune and less interpretable than a clear, thresholded gate.

Our design avoids the common pitfalls of these more complex techniques while enabling fast, stable, and theoretically-grounded inference.

### C.5 Justification for Methodological Design Choices

This section provides additional justification for two main design choices: (1) the use of a fixed-weight loss combination for multi-task training, and (2) the selection of specific model backbones for our AHA framework.

#### C.5.1 On the Use of Fixed-Weight Multi-Task Training

Using a fixed, weighted sum of multiple losses is not only a de-facto standard in large-scale pre-training (e.g., BERT [64] adds Masked LM and NSP losses) but also a classic scalarization strategy in multi-objective optimization. When each task loss $L_i(\theta)$ is well-behaved, optimizing

$$L_0(\theta) = \sum_{i=1}^{T} w_i \, L_i(\theta), \quad w_i > 0, \ \sum_{i=1}^{T} w_i = 1$$

converges to a point on the convex Pareto front [65], guaranteeing that no objective can be improved without degrading another.

Recent empirical studies have rigorously compared simple fixed-weight scalarization against more complex, specialized multi-task optimizers (SMTOs). These studies show that with appropriate normalization and tuning, scalarization can match or even surpass these dynamic methods on diverse benchmarks [66, 67]. The primary advantages of this approach are its stability and scalability, as it avoids the significant per step computational overhead inherent in dynamic re-weighting schemes. Standard regularization techniques like weight decay and dropout also help mitigate conflicting gradients, reducing the need for more complex optimizers. Thus, our choice is grounded in a strong foundation of theoretical guarantees and practical evidence.

#### C.5.2 On the Selection of Model Backbones

The selection of the Qwen2 [29] and SigLIP [27] backbones was based on a thorough review of high performing, open-source multimodal models at the time of this work.

**Visual Encoder (SigLIP):** We selected SigLIP as it has been shown to offer competitive or superior generalization compared to CLIP, particularly at the smaller batch sizes that are characteristic of our online, per-frame processing setup.

**Language Backbone (Qwen2):** For the language backbone, we adopted the LLaVA-OneVision architecture based on Qwen2. The distilled 7B variant of this model demonstrates SOTA performance while remaining lightweight enough for our framework.

Both Qwen2 and SigLIP are widely used and validated in concurrent streaming vision-language literature [13], showcasing their competitiveness and broad community adoption. While testing on additional backbones is an important direction for future work, these selections represent a well-grounded starting point for establishing the AHA framework.

## D Dataset Curation for Informativeness and Language Modeling Heads

This appendix provides further details on the creation of ground truth labels used to supervise the informativeness and auxiliary LM heads of the AHA framework. For both heads, we leverage existing video-language datasets with segment-level captions, specifically the human-annotated subset of

**Shot2Story** [35] and longer procedural videos from **COIN** [36]. The methodology for generating supervision signals follows the same strategies employed in the streaming framework MMDuet [13].

### D.1 Supervision for the Informativeness Head

The informativeness head in AHA is trained to predict whether the current video frame $f_t$ introduces new information relative to the preceding context. Our approach for generating ground-truth labels for this task is a heuristic adopted directly from established work in streaming Video-LLMs [13]. The original motivation in that context was to train a model that knows *when to speak* during a continuous video stream, generating a response only after acquiring sufficient context but before the moment becomes stale.

We adapt this principle to derive binary labels ($i_t \in \{0, 1\}$) as follows:

1. **Segment Identification:** We utilize video segments with corresponding human-generated captions from the Shot2Story and COIN datasets.

2. **Point of Sufficient Understanding:** For each segment, we simulate a point where enough information has been seen to describe it. This point is randomly sampled to occur between 50% and 75% of the segment's duration.

3. **Label Assignment:** Frames from the 50% mark up to the "point of sufficient understanding" are labeled as informative ($i_t = 1$). All other frames in the segment (before 50% or after the point) are labeled non-informative ($i_t = 0$).

The underlying intuition is that initial frames may lack context, while frames after understanding is achieved are redundant. We hypothesize that this signal, which marks the accumulation of new information, correlates with highlight-worthy moments in an OHD setting. This is a hypothesis supported by our strong ablation results (Table 3).

**Decoupling Informativeness from Relevance.** A key design choice in AHA is the explicit decoupling of the informativeness head from the relevance head. While related, informational novelty (informativeness) and task-importance (relevance) are distinct concepts. To validate that our model learns these different signals and that the concept of informativeness generalizes beyond the procedural videos used for training, we conducted a qualitative analysis on the unconstrained, real-world SCOUT robotics video. Given the task objective ($\mathcal{Q}$) "what objects are in this room?", we observed the following distinct behaviors:

- **High Informativeness, Low Relevance:** When the robot enters a dark room, the drastic scene change correctly triggers a high informativeness score due to visual novelty. However, with no task-relevant objects visible, the relevance score remains low.

- **Low Informativeness, High Relevance:** Conversely, if a task-relevant "calendar" is visible from afar, both scores are initially high. As the robot moves closer, the informativeness score drops because the visual context is no longer novel. The relevance score, however, spikes as the calendar becomes clearly identifiable, confirming its task importance.

- **Correlated Signals:** The scores often peak in unison when the robot enters a new area and immediately encounters a task-relevant object (e.g., "a shovel"). Even in these cases, the relevance head typically produces a higher peak, correctly prioritizing the task-specific discovery over the general novelty of the scene.

This analysis confirms that our decoupled design is effective. The informativeness head successfully captures visual novelty in unconstrained environments, while the relevance head remains focused on the specific task objective, allowing a more robust understanding of the video stream.

### D.2 Supervision for the Auxiliary Language Modeling (LM) Head

To enrich the semantic quality of the hidden representations ($h_t$) learned by AHA, an auxiliary LM head is incorporated. This head is trained using the same dense captioning annotations from the Shot2Story and COIN datasets as the informativeness head.

The training process is as follows:

1. At randomly selected timesteps $t$ during training, the LM head is tasked with generating a short, descriptive caption for the current visual context encapsulated by frame $f_t$.

2. This generation is conditioned on the prior context available to the model (i.e., preceding visual tokens and the fixed task prompt $\mathcal{Q}$ and system prompt $\mathcal{S}$).

3. The supervision is provided via a standard cross-entropy loss for next-token prediction against the ground truth human-annotated captions corresponding to that segment (Eq. 6).

It is crucial to reiterate a key design choice for AHA that distinguishes its use of the LM head from some interactive VideoLLMs like MMDuet:

- The captions generated by AHA's LM head during training are *not* re-injected into the model's input context.

- Similarly, the LM head is typically not used during inference for the primary task of highlight detection (unless its hidden states are implicitly part of $h_t$).

This approach strictly preserves AHA's unidirectional, non-dialogue streaming behavior, ensuring it functions purely as a continuous scorer of video frames against a static task objective. The LM task serves solely as a mechanism to improve the overall quality, alignment, and semantic richness of the hidden state representations ($h_t$) from which the primary highlight detection scores (relevance, informativeness, uncertainty) are derived.

## E  Supplementary Quality Dropout Details

This appendix section offers supplementary details and justifications for certain design choices introduced in the main paper's HIHD methodology (Section 3.3) and the robustness experiments (Section 4.2). These elaborations are provided here to expand upon descriptions that were necessarily concise in the main text.

### E.1  Video Quality Dropout for Robustness Enhancement

To improve the robustness of AHA against visual artifacts and degradations commonly encountered in real-world video streams, we incorporate a video quality dropout mechanism during the training data preparation phase. As described in Section 3.3, for each video in our HIHD data, 5–20% of its duration is randomly selected for augmentation. Frames within these selected segments undergo one of several random perturbation types, detailed below. This process helps the model learn to maintain performance despite noisy or imperfect visual input [41]. Let $f(x, y)$ denote the pixel values at coordinates $(x, y)$ of an input frame $f$.

- **Quality Degradation:** This simulates general compression artifacts and loss of detail. The frame $f$ is first downscaled to a fixed low resolution (e.g., $H' \times W' = 64 \times 64$) using bilinear interpolation, denoted as $D(f; H', W')$. This downscaled frame $f_{small} = D(f; H', W')$ is then upscaled back to the original dimensions $H \times W$ using nearest-neighbor interpolation, $U(f_{small}; H, W)$, to preserve blockiness. Finally, a Gaussian blur $G_{\sigma,k}$ with kernel size $k$ (e.g., $(5, 5)$) and standard deviation $\sigma$ (e.g., 0) is applied.

$$f' = G_{\sigma,k}(U(D(f; H', W'); H, W))$$

- **Block Noise:** This simulates digital transmission errors. The frame $f$ is notionally divided into non-overlapping blocks $b_{ij}$ of size $B_s \times B_s$ (e.g., $32 \times 32$). A fixed random noise pattern $N \in [0, R_{max}]^{B_s \times B_s \times C}$ (e.g., $R_{max} = 49$, representing noise intensity up to 49 for an 8-bit channel) is generated. For each block $b_{ij}$, it is replaced by $N$ with a probability $p_{noise}$ (e.g., $p_{noise} = 0.1$). Let $m_{ij} \sim \text{Bernoulli}(p_{noise})$ be a random variable for each block.

$$f'(x, y) = \begin{cases} N(x \bmod B_s, y \bmod B_s) & \text{if } m_{\lfloor x/B_s \rfloor, \lfloor y/B_s \rfloor} = 1 \\ f(x, y) & \text{if } m_{\lfloor x/B_s \rfloor, \lfloor y/B_s \rfloor} = 0 \end{cases}$$

This operation is applied over all pixel coordinates $(x, y)$.

- **Color Banding:** This simulates reduced color depth, leading to visible bands in color gradients. Each pixel channel value $P \in [0, 255]$ in the frame $f$ is quantized using a quantization factor $Q$ (e.g., $Q = 64$)

$$f'(x,y)_c = \left\lfloor \frac{f(x,y)_c}{Q} \right\rfloor \cdot Q$$

for each channel $c$.

- **Blackout:** This simulates a complete loss of signal. All pixel values in the frame are set to zero

$$f'(x,y)_c = 0$$

for all channels $c$ and coordinates $(x, y)$.

During training, one of these dropout types is randomly chosen and applied to frames within the selected 5–20% dropout segments of a video. To ensure the model always has some visual context for making predictions and to prevent scenarios where all frames in its immediate processing window might be entirely obscured (e.g., by consecutive 'blackout' frames), we limit consecutive blackout augmentations to a maximum of $X$ frames (e.g., $X = 5$ at 1fps). If a frame is scheduled for blackout beyond this limit, a milder form of degradation (such as quality reduction or color banding) or no augmentation is applied instead for that specific frame, after which the possibility of blackout frames resumes. This prevents the model from being trained on entirely uninformative sequences over extended periods, which could hinder learning. Corresponding dropout masks are also generated alongside the HIHD data. This augmentation strategy is critical for preparing AHA to handle the unpredictable visual quality often present in real-world online video streams. A visual representation of these methods are shown in Figure 4

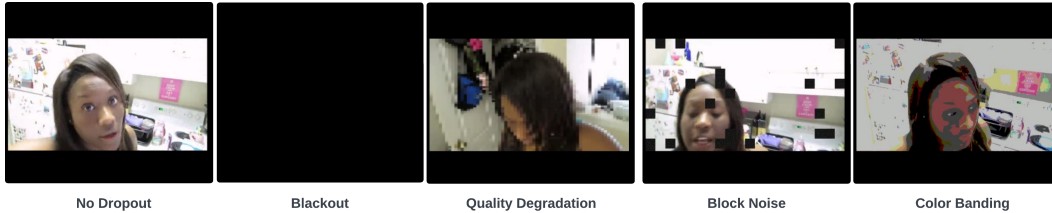

|  No Dropout | Blackout | Quality Degradation | Block Noise | Color Banding |

Figure 4: Visualization of dropout modes used during training to simulate real-world sensor degradation and robustness challenges. From left to right: No Dropout (clean input), Blackout (entire frame lost), Quality Degradation (strong downsampling and blurring), Block Noise (random black patches), and Color Banding (aggressive color quantization). These corruptions are applied to random segments of video to improve model robustness to visual noise. [5]

### E.2 Quality Dropout Results: Kendall $\tau$ and Spearman $\rho$

This appendix presents supplementary results for the robustness analysis on the TVSum dataset, evaluated using Kendall's $\tau$ and Spearman's $\rho$ rank correlation coefficients. These metrics offer an alternative perspective on the models' ability to maintain ranking performance under various video degradations. As mentioned in the main text, these rank correlation metrics generally showed minor variations across conditions for our model, AHA (Ours), and reinforced the overall conclusions regarding robustness. The detailed scores are provided in Table 13.

The data in Table 13 for AHA (Ours) shows that while there are some fluctuations, particularly with the blackout degradation, the rank correlation scores remain relatively stable across several milder degradation types, supporting the conclusions discussed in the main paper.

---

[5]Screenshot from the YouTube video "Vlog #509 I'M A PUPPY DOG GROOMER! September 13, 2014" licensed under Creative Commons Attribution 3.0 (CC BY 3.0) via YouTube. Source: https://www.youtube.com/watch?v=Bhxk-O1Y7Ho

Table 13: Robustness to video degradations on TVSum (Kendall $\tau$ / Spearman $\rho$). Degradations applied to 20% of frames.

| Model | Clean | +ColorBanding | +BlockNoise | +Quality | +Blackout |
|---|---|---|---|---|---|
| AHA (Ours) | 0.28/0.40 | 0.28/0.40 | 0.29/0.40 | 0.28/0.39 | 0.24/0.34 |

# F  Memory Management for Streaming OHD: From SinkCache to Dynamic SinkCache

This section details the evolution of our memory management strategy, from adopting the standard SinkCache mechanism to developing our novel, higher-performing Dynamic SinkCache.

## F.1  Standard SinkCache as a Hybrid Memory Baseline

To manage the challenge of unbounded KV cache growth when processing continuous video streams, our initial framework adopted the SinkCache mechanism [21]. This hybrid memory strategy ensures constant memory usage by maintaining two components: a fixed set of initial **Sink Tokens** ($k_s$) for long-term context (like the task objective) and a sliding window of **Recent Tokens** ($k_{t-n:t}$) for short-term context. Any tokens outside this combined memory are evicted. An illustration of this standard memory structure is provided in Figure 5.

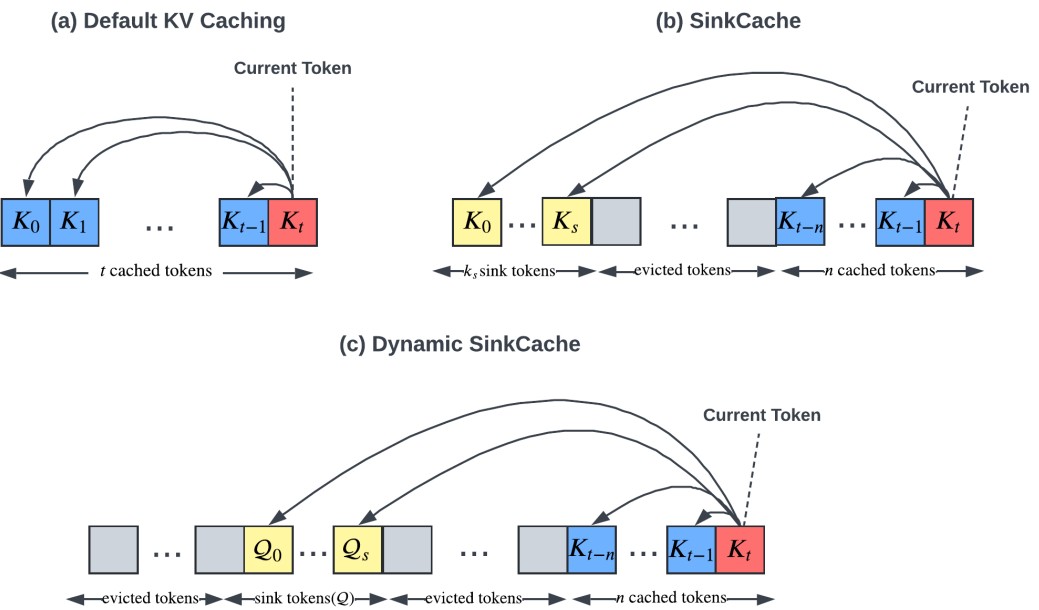

Figure 5: Comparison of memory structures: (a) Default KV Caching, which increases memory linearly. (b) SinkCache, where the current token attends to a hybrid memory comprising a fixed set of initial sink tokens and a sliding window of recent tokens. (c) Dynamic SinkCache, where the sink is dynamically constructed to contain only the task objective ($\mathcal{Q}$) tokens, combined with a sliding window of recent tokens. This preserves long-term context while maintaining constant memory.

## F.2  Justification of Sliding Window Size (n=2048)

While the Dynamic SinkCache creates a targeted sink for the task objective, the size of the sliding window for recent visual tokens ($n$) remains a key hyperparameter. A larger window provides more short-term context at the cost of higher memory and computational overhead, while a smaller window is more efficient but may lose critical immediate context.

To find an optimal balance, we conducted an ablation study on the standard SinkCache over various sink ($|k_s|$) and window ($n$) sizes, with the results summarized in Table 14.

Table 14: Ablation on TVSum for Standard SinkCache sink ($|k_s|$) and window ($n$) sizes.

| SinkCache Config. ($|k_s|, n$) | Top-5 mAP | Spearman's $\rho$ | Kendall's $\tau$ |
|---|---|---|---|
| (32, 2048) | **92.6** | 0.401 | 0.280 |
| (16, 2048) | 92.0 | 0.295 | 0.203 |
| (32, 1024) | 90.1 | 0.412 | 0.287 |
| (40, 2560) | 89.4 | 0.298 | 0.205 |
| (16, 1024) | 89.1 | 0.359 | 0.247 |
| (16, 512) | 84.0 | 0.216 | 0.145 |

The results show that a window size of $n = 2048$ paired with $|k_s| = 32$ sink tokens achieved the highest mAP. While performance degrades gracefully with smaller windows (e.g., $n = 1024$ still achieves a strong 90.1 mAP), $n = 2048$ proved to be the optimal configuration. We therefore adopted this window size for our final Dynamic SinkCache implementation, as it provides the best performance by capturing sufficient recent visual context for the OHD task.

## F.3 Dynamic SinkCache: A Task-Focused Improvement

We hypothesized that the standard SinkCache's method of using the *first few tokens* as a generic sink was suboptimal. These initial tokens capture the system prompt, task objective, and sometimes the first few video frames. We proposed that a more targeted sink, containing *only* the essential task information, would provide a cleaner and more effective long-term memory.

This led to our novel approach, the **Dynamic SinkCache**. Instead of using the first $s$ tokens of the sequence, this mechanism dynamically constructs the sink to contain exclusively the natural language **task objective tokens** ($\mathcal{Q}$). This ensures that the model's long-term memory is persistently and exclusively focused on its primary goal, preventing it from being diluted by less relevant initial context.

## F.4 Comparative Analysis of Memory Mechanisms

To validate our final design choice (Dynamic SinkCache) and demonstrate the necessity of a hybrid, task-focused memory system, we conducted a comprehensive ablation study on TVSum. We compared five different memory management strategies, which are detailed below. The results are summarized in Table 15.

**Baseline Mechanisms.** We first evaluated two simple, non-hybrid baselines. A **Sliding Window Only** approach, which retains only the most recent visual tokens, performed poorly (69.5 mAP) because it eventually discards and forgets the long-term task objective. Conversely, a **Static Window Only** approach, which uses only the initial tokens as context, performed even worse (63.2 mAP) as it completely fails to adapt to new visual events in the video stream.

**Unbounded KV Cache.** As a practical upper-bound, a standard unbounded KV cache that retains all previous tokens achieved a strong 91.7 mAP. However, this method is impractical for real-world deployment, as its linear memory growth consistently leads to out-of-memory (OOM) errors on the long videos common in OHD tasks.

**Standard SinkCache.** The standard SinkCache [21], which combines a generic sink of the initial sequence tokens with a sliding window, proved to be a highly effective hybrid baseline. It achieved 92.6 mAP, outperforming the impractical unbounded cache while maintaining a constant memory footprint.

**Dynamic SinkCache (Ours).** Our proposed method achieves the highest score of 93.0 mAP. By dynamically constructing the sink to contain exclusively the natural language task objective, it creates a more targeted and efficient long-term memory. This confirms our hypothesis that a task-focused sink provides the optimal mechanism for context retention in OHD.

Table 15: Ablation study of memory mechanisms on TVSum (Top-5 mAP).

| Memory Mechanism | Top-5 mAP | Notes |
|---|---|---|
| Sliding Window Only | 69.5 | Fails to retain the long-term task objective. |
| Static Window Only | 63.2 | Fails to adapt to new visual events. |
| Unbounded KV Cache | 91.7 | Strong performance but impractical (causes OOM). |
| Standard SinkCache | 92.6 | Effective hybrid memory, strong baseline. |
| **Dynamic SinkCache (Ours)** | **93.0** | **Optimal performance with task-focused sink.** |

### F.5 Limitation and Design Trade-off

A key consideration of the Dynamic SinkCache is the trade-off between the sink size (determined by the task objective length) and the sliding window size for recent visual tokens. Our implementation assumes a fixed total memory capacity. The strong performance in our experiments is partly due to the concise nature of the task objectives in benchmarks like TVSum, which occupy a small, reasonable portion of the cache (~45 tokens), leaving enough capacity for the sliding window tokens.

However, this showcases a limitation: the model's performance could degrade catastrophically if presented with an exceptionally long natural language objective. If a task description were long enough to consume the entire memory budget, the sliding window for recent visual tokens would be eliminated. In this scenario, the model would retain the task but lose all short-term visual context, rendering it unable to perform the OHD task. This trade-off underscores the need for future work in developing more adaptive memory allocation schemes that can handle tasks with highly variable objective lengths.

## G Supplementary Details for Real-World Robotic Evaluation on SCOUT Video

This appendix provides additional details that supplement the evaluation of AHA on the SCOUT video presented in Section 4.3.

### G.1 Additional SCOUT Video Characteristics

Beyond the general description of the SCOUT video [16] in the main text (long-horizon, continuous footage, degraded quality, sparse events), the video present further specific challenges relevant to real-world deployment. These include:

- **Severe Visual Degradations:** The footage contains periods of near-complete *blackout* (e.g., when the robot navigates very dark areas) and intermittent *signal static*, in addition to the warping mentioned in the main text.

- **Domain and Visual Noise:** The dataset is characterized by a significant *domain shift toward indoor navigation* compared to common web datasets, and often contains *high visual noise* and *unpredictable robot motion*.

### G.2 Ground Truth Annotation Specifics for SCOUT Qualitative Analysis

For the 8-minute qualitative analysis discussed in Section 4.3, the ground truth events (i.e. the peaks matching events) were identified by the authors of this paper. This process involved a visual comparison of AHA's highlight detection outputs (specifically, the predicted peaks after Savitzky-Golay smoothing [42]) against:

1. Moments in the video where the robot was observed to be stationary, often indicating task completion or observation of a point of interest.

2. Timestamps corresponding to human-issued navigation instructions for the robot, as documented in the official SCOUT transcripts [16].

This refined how "meaningful actions" were correlated with AHA's predictions.

### G.3 Nuanced Analysis of Predicted Peaks in SCOUT Evaluation

Section 4.3 reports that 16 of 18 predicted peaks from AHA aligned with human-issued commands or meaningful actions. Further details on the remaining two peaks are as follows:

- One peak corresponded to AHA identifying an object of interest (based on mission context from SCOUT annotations) while the robot was still in motion executing a prior command.

- The other peak did not strongly correlate with a new command or a clearly defined object of interest from the mission logs for that segment. This might represent model-perceived visual saliency not directly tied to the high-level task commands, or a potential false positive.

While this analysis is preliminary and conducted on a single SCOUT video, the results encourage continued exploration of this domain and analysis on additional videos.

### G.4 Expanded Implications and Future Work for Robotics Applications

The application of AHA to the SCOUT video suggests further implications for robotics beyond those outlined in Section 4.3:

- **Targeted Operator Alerting:** AHA could potentially alert a human operator specifically if the robot perceives an object of interest that the operator might have missed, particularly if it's an unexpected finding or occurs while the robot is still executing a previous command.

- **Synergy with Human-Robot Dialogue Systems:** Combining AHA's perceptual salience with intent-aware dialogue systems, such as those explored in prior SCOUT work [68], could:

  - Help flag video segments associated with human commands where perceptual ambiguity (e.g., unusual saliency detected by AHA that is not aligned with the stated task) might indicate potential misunderstandings or execution challenges.
  - Assist in grounding conversational references (e.g., a human asking "what was that interesting thing we just passed?") to specific video segments highlighted by AHA.

- **Input for Multimodal Reasoning:** AHA's real-time, frame-level salience scores can serve as a valuable input signal for more comprehensive multimodal reasoning frameworks, helping to focus computational resources on the most pertinent segments of continuous video data.

## H   Query Templates for Task Objective Generation

For the Human Intuition Highlight Dataset (HIHD), synthetic task objectives ($\mathcal{Q}$) are generated by programmatically transforming video titles using the following templates. Given a video title represented as '[STRING]', a query is randomly selected from this list:

```
query_templates = [
    "[STRING]", # Repeating the title itself can serve as a direct query
    "What segment of the video addresses the topic '[STRING]'?",
    "At what timestamp can I find information about '[STRING]' in the video?",
    "Can you highlight the section of the video that pertains to '[STRING]'?",
    "Which moments in the video discuss '[STRING]' in detail?",
    "Identify the parts that mention '[STRING]'.",
    "Where in the video is '[STRING]' demonstrated or explained?",
    "What parts are relevant to the concept of '[STRING]'?",
    "Which clips in the video relate to the query '[STRING]'?",
    "Can you point out the video segments that cover '[STRING]'?",
    "What are the key timestamps in the video for the topic '[STRING]'?"
]
```

This process generates a diverse set of queries for each video, enabling task-conditioned supervision.

# I  Future Work: Supervised Learning and Validation with MultiVENT-G

While our work establishes a strong empirical baseline for OHD, two key areas for future improvement are the unsupervised nature of our uncertainty estimation and the inherent biases of our large-scale HIHD dataset. Our current approach to uncertainty is unsupervised due to the profound difficulty of obtaining ground-truth confidence labels at scale. Similarly, HIHD relies on YouTube's "Most Replayed" data, a high throughput but imperfect proxy for importance that can be influenced by engagement driven biases like clickbait.

A promising path to address these limitations involves leveraging the recently released MultiVENT-G dataset [44]. Focused on high stakes disaster events, MultiVENT-G provides two critical features missing from typical highlight detection datasets: (1) dense, frame-level event role annotations by human experts, and (2) human-annotated confidence scores (1-5 scale) for these annotations. This dataset offers a unique opportunity to advance our work in three key directions:

1. **Towards Supervised Uncertainty:** The annotator confidence scores can be transformed into ground-truth uncertainty labels (e.g., 5/5 confidence maps to low uncertainty). This would allow training our uncertainty head with a direct supervised loss, moving beyond the current unsupervised NLL objective. This is a critical step towards improving the interpretability and calibration of our model's confidence estimates, which is essential for the safety-critical applications we target.

2. **Mitigation of Dataset Bias:** MultiVENT-G's expert defined event labels (e.g., '"EMERGENCY-RESPONSE") can serve as a gold-standard signal of true relevance. This allows for future work in calibration and debiasing, where our model, pre-trained on the large-scale HIHD, can be fine-tuned on MultiVENT-G. This process would help correct for systemic biases learned from the raw replay scores, yielding a model that is more faithful to expert defined importance.

3. **Validation in High Stakes Domains:** By providing task-aligned ground truth for disaster events, MultiVENT-G allows for rigorous validation of our model in the exact high stakes scenarios for which it is designed. This ensures that the relevance and uncertainty estimates converge toward what human experts deem critical in real-world applications.

Despite its potential, integrating MultiVENT-G presents three primary challenges: Scale (MultiVENT-G's ~1.2k videos vs. HIHD's ~23k), Generalization (its specific ontology may constrain the learned representations), and Subjectivity (labels are from a small team of annotators). Our future work will focus on developing methods to address these challenges, aiming to create a model that is not only scalable but also robust, well-calibrated, and grounded in expert knowledge.

