# OpenReview forum: "Aha! - Predicting What Matters Next: Online Highlight Detection Without Looking Ahead"
_NeurIPS.cc/2025/Conference — NeurIPS 2025 poster_

### Official Review · Reviewer_Zkgi · 2025-06-04

**Clarity:** 3
**Significance:** 3
**Originality:** 3
**Rating:** 5
**Confidence:** 3

**Summary:**

This paper presents an online highlight detection framework that processes continuous video streams in real-time without relying on future frames. Unlike existing methods that require the entire video for analysis, this framework works frame by frame, connecting video content with a natural language task description. It uses a multimodal vision-language model and lightweight, decoupled heads trained on a large, curated dataset of human-centric video labels. To ensure scalability, it uses a memory-efficient technique called SinkCache, allowing it to handle very long videos with constant memory usage. An empirical study indicates that the proposed framework outperforms existing video highlight detection methods and demonstrates strong potential for real-world, time-sensitive applications like robotics and surveillance.

**Questions:**

Q1. How do you capture long-horizon video features when using fixed-length windows during video analysis?

Q2. How could the window size affect the framework's performance? Is it possible to conduct experiments on various window sizes?

Q3. When analyzing real-time video streams, is there a report on the framework's latency? What computing resources (e.g., GPUs) are required for real-time video analysis?

**Ethical Concerns:**

["NO or VERY MINOR ethics concerns only"]

**Final Justification:**

I have read the rebuttal and think that the authors have addressed all the concerns I have. Hence, I will keep my score.

**Limitations:**

L1. The paper includes an uncertainty prediction head for uncertainty estimation. However, the projection head is trained without explicit uncertainty annotations (e.g., a calibration dataset with ground truth labels). This limits the interpretability and calibration of the uncertainty outputs, which could be critical in safety-sensitive domains.

L2. This work is heavily relying on architecture design and empirical studies, but lacks theoretical support. It would be beneficial to include a discussion on formal guarantees on the projection heads' outputs, or some theoretical analysis on loss convergence and hyperparameter selections.

**Paper Formatting Concerns:**

There are no formatting concerns.

**Quality:**

3

**Strengths And Weaknesses:**

**Strengths**

The paper introduces a framework that can process video streams in real time without processing future frames. This feature allows the framework to process ideally infinite-length videos with constant memory usage, which significantly reduces computation overhead.

Empirical studies show that the framework outperforms existing methods in highlight detection on benchmark datasets, showcasing its effectiveness with relatively small computation overhead.

The framework has potential in other streaming video understanding tasks beyond highlight detection, showcasing its broad applicability.

**Weaknesses**

As mentioned in the limitation section, while it is effective in the inference stage, the training cost is relatively high.

The uncertainty estimation lacks formal or theoretical support.

*Note to the authors*: Please respond to the **Questions** and **Limitations** sections during the rebuttal period (instead of responding to this section).

---

> ### Author Rebuttal · Authors · 2025-07-30
>
> We thank the reviewer for their positive assessment and for highlighting our framework's real-time capabilities and strong empirical performance. We are grateful for your insightful questions and feedback on the paper's limitations, which we address below.
>
> ## Q1 (Capturing Long-Horizon Features)
> Our framework captures long-horizon features via the **SinkCache** mechanism. This is a hybrid approach that separates memory into two critical components: 1) a persistent **sink** that retains the initial, global task-conditioning tokens (i.e., the natural language query), acting as a long-term memory for the mission objective, and 2) a fixed-size **sliding window** of recent tokens that processes the immediate visual context. Our new ablation experiment (detailed in our response to reviewer 79jx - R1) confirm this design is critical: a sliding window only approach fails because it forgets the task (achieving only a 69.5 mAP in comparison to our reported 92.6 mAP), proving that the sink is what retains the essential long-horizon information.
>
> ## Q2 (Impact of Window Size)
> We thank the reviewer for raising this point and apologize that our notation in Table 3 on page 8 was not sufficiently clear. We will revise the table caption in the revised paper version to explicitly state that $|k_s|$ denotes the number of persistent sink tokens and $n$ represents the size of the sliding window for recent tokens, which directly corresponds to the "window size" in question. Table 3 (right) details our ablation study over these parameters. Our results show that while our optimal configuration using a window size of $n=2048$ achieves our best mAP of 92.6, the framework's performance degrades gracefully as the window size is reduced. For example, halving the window to $n=1024$ maintains a strong mAP of 90.1, and even a minimal configuration with $n=512$ achieves an 84.0 mAP, demonstrating the model's robustness to smaller context windows. This analysis confirms that our chosen configuration provides an excellent balance between performance and memory efficiency. Should the reviewer find a more extensive analysis beneficial, we would be happy to report results on a wider range of window sizes.
>
> ## Q3 (On Latency and Compute Resources)
>
> We have conducted a detailed performance analysis of our framework on a 1062 second video (~17 minutes) using two NVIDIA A6000 GPUs. The system achieved a sustained throughput of 1 frame per second (FPS), demonstrating high efficiency with 100% peak GPU utilization and 90% peak memory controller utilization. During this process, the framework consumed a peak of 30.49 GB of VRAM across both GPUs and operated well within safe thermal limits at a peak temperature of 65°C, all while maintaining a minimal system RAM footprint of 3.66 GB. While this establishes a strong performance baseline, the 1 FPS rate means that in a live scenario, depending on the GPU, the system would fall behind the incoming video feed (drift), requiring frame-skipping logic for real-time deployment. Given that our framework already leverages the available compute effectively, it is a strong candidate for such optimizations to achieve the higher throughput needed for live applications. We thank the reviewer for bringing up a practical question and will expand Appendix A with this information, and include a summary and pointer to the Appendix in the main paper.
>
> ## L1 (On Uncertainty Estimation without Ground-Truth Labels)
>
>
> **Justification for Unsupervised Training:** This is a critical point, and the reviewer is correct that the uncertainty head is trained without explicit ground-truth labels, which we note as a key limitation in section 5. This is due to the profound difficulty in obtaining such labels; a "highlight" is subjective, and an annotator's confidence is even more so, making it an unsolved challenge to collect these labels at scale.
>
> Regarding the lack of ground-truth labels, which we identify as a key limitation, we adopted a principled, unsupervised approach using a Gaussian Negative Log-Likelihood (NLL) loss[1]. This leads directly to the intuition for the variance diversity loss.
>
> The underlying intuition for encouraging variance diversity in Equations 5a and 5b is to **prevent mode collapse**. A known issue with this type of unsupervised uncertainty modeling is that the model can learn a degenerate solution by predicting a single, uninformatively high variance for all frames to minimize the NLL loss[2]. The diversity regularizer, $\mathcal{L}_{\text{div}}$, directly counteracts this by penalizing a low standard deviation of the predicted log-variances across a batch. This encourages the model to produce a dynamic and meaningful range of uncertainty values, allowing it to distinguish between predictable and ambiguous moments in a video stream. We will supplement our discussion in the limitations with this additional information.
>
> **Future Work: Towards Supervised Uncertainty with MultiVENT-G:** We are actively exploring datasets that could provide labels for our supervised uncertainty learning in future work. One promising direction we have identified for *supervised* uncertainty-aware OHD involves leveraging the **MultiVENT-G** dataset[3], which focuses on complex, real-world events like disasters and provides a key feature missing in typical HD datasets: **human-annotated confidence scores**. Annotators rated their confidence on a 1-5 scale that a visual entity pertained to an event role.
>
> This dataset offers three key advantages:
> - **Map Confidence to Uncertainty:** The confidence scores can be directly transformed into ground-truth uncertainty labels (e.g., a 5/5 confidence rating corresponds to low uncertainty).
> - **Generate a New Training Objective:** With these new labels, the uncertainty head could be trained directly with a supervised loss to address the critical issues of interpretability and calibration.
> - **Validate in High-Stakes Domains:** Since MultiVENT-G is focused on disaster events, it allows for the validation of uncertainty-aware models in the exact high-stakes applications our paper targets.
>
> However, there are currently three primary challenges with this approach: (1) **Scale**, as MultiVENT-G contains only 1,168 videos compared to the ~22.5k in our HIHD dataset; (2) **Generalization**, as training on MultiVENT-G's specific ontology may constrain the learned uncertainty; and (3) **Subjectivity**, as the labels are derived from a small team of annotators. We will create a new section in the appendix clearly outlining this promising direction and its associated challenges in Appendix J for our revised paper.
>
> **Addressing Safety-Critical Applications and Responsible AI:** Finally, in response to the reviewer's important point on safety-sensitive applications, we acknowledge this is currently limited by available data. A core goal in releasing our **HIHD dataset** is to provide a large-scale resource that can help the responsible AI community develop methods for identifying ground truth and safety critical labels. We are actively exploring collaborations to tackle these issues and, based on the reviewers feedback, will move a stronger summary of this future work to the main paper. Thank you for bringing this important topic to our attention.
>
> ## L2 (On Lack of Theoretical Support)
>
> We agree that our work is primarily empirical, focusing on introducing a new task (OHD), a novel framework, and a large-scale dataset. While we have not derived new formal guarantees, the architectural and training principles we use are grounded in existing literature. Specifically, our multi-head training approach uses a fixed weighted sum of losses. This is a scalarization strategy in multi-objective optimization that is theoretically guaranteed to find a Pareto-optimal solution[4]. It is also an empirically validated standard in large-scale deep learning.
>
> We have now strengthened the paper's connection to this established literature and explicitly call out the need for future theoretical analysis of OHD systems as an important research direction. We thank the reviewer for encouraging us to make these connections clearer.
>
> [1]: Nix, David A., and Andreas S. Weigend. "Estimating the mean and variance of the target probability distribution."
>
> [2]: Seitzer, Maximilian, et al. "On the pitfalls of heteroscedastic uncertainty estimation with probabilistic neural networks."
>
> [3]: Sanders et al., "Grounding Partially-Defined Events in Multimodal Data."
>
> [4]: Miettinen, Kaisa. "Nonlinear multiobjective optimization."

---

> > ### Comment · Reviewer_Zkgi · 2025-08-05
> >
> > The rebuttal has addressed all my questions and concerns. Please integrate the results presented in the rebuttal into the paper during revision.
> > I will keep my current score and vote for acceptance.

---

> > > ### Author Response · Authors · 2025-08-05
> > >
> > > We are very grateful for your positive feedback and for confirming your strong support for our paper. We are happy to hear that our rebuttal successfully addressed all your concerns. We confirm that we will integrate all the results and discussions from the rebuttal into the final manuscript, as you requested.
> > >
> > > Thank you again for your time and for helping us improve our work!

---

### Official Review · Reviewer_NtaH · 2025-06-27

**Clarity:** 3
**Significance:** 3
**Originality:** 3
**Rating:** 5
**Confidence:** 3

**Summary:**

This paper proposes AHA, an autoregressive model for online highlight detection in video streams without relying on future frames. The framework integrates a multimodal transformer architecture enhanced with multiple heads for relevance, informativeness, and uncertainty prediction.  AHA incorporates the SinkCache mechanism, a approach enabling constant-memory inference for handling continuous video streams. The authors further introduce a new large-scale Human Intuition Highlight Dataset (HIHD) to facilitate model training. Experiments illustrate that AHA outperforms current state-of-the-art methods.

**Questions:**

1. How sensitive is the model's performance to variations in the chosen loss weights?
2. How robustly does SinkCache retain good context over extended video durations (e.g., hours-long streams)?
3. How reliable is AHA when provided ambiguous or irrelevant task conditioning?

**Ethical Concerns:**

["NO or VERY MINOR ethics concerns only"]

**Final Justification:**

The authors' rebuttal has addressed all my concerns and questions. After reading the rebuttal and other reviewers' comments, I decide to keep my score and recommend accept.

**Limitations:**

Yes

**Paper Formatting Concerns:**

No formatting concerns.

**Quality:**

3

**Strengths And Weaknesses:**

Strengths:

1. The proposed AHA model advances in real-time video understanding without the dependence on future frame information, thereby aligning well with practical streaming scenarios.
2. The proposed method demonstrates strong performance under causal conditions on benchmark datsets.
3. The new HIHD dataset provides contributions for future research in OHD.

Weaknesses:
1. Although justified through ablation studies, the multi-head training and fixed weighting strategy could benefit from more theoretical or automated justification.
2. The practical advantages of uncertainty modeling are unclear.
3. In additional tasks, the model exhibits redundant or repetitive predictions.

---

> ### Author Rebuttal · Authors · 2025-07-30
>
> We sincerely thank the reviewer for their positive assessment and recognition of our work's strengths, and for providing constructive feedback with insightful questions that help us further improve the paper.
>
> ## Q1, W1 (Justification for Multi-Head Training and Weight Sensitivity)
> The reviewer raised an excellent point about providing more justification for our fixed-weight, multi-head training strategy beyond the initial ablation studies. We provide our response here, and will add it to the main paper.
>
> **Theoretical and Empirical Justification:** Our choice follows established practices in prior work. Using a fixed weighted sum of multiple losses is a common approach in large-scale pre-training (e.g., BERT[1]) and aligns with the scalarization strategy in multi-objective optimization. When each task loss $L_i(\theta)$ is well-behaved, optimizing $$L_0(\theta) = \sum_{i=1}^T w_i\,L_i(\theta), \quad w_i>0,\;\sum_{i=1}^T w_i=1$$
>
> converges to a point on the convex Pareto front[2], guaranteeing that no objective can be improved without degrading another. While we demonstrated theoretically this would represent trade-offs, we agree an empirical sensitivity analysis is an important analysis. Due to the high computational cost of retraining our model (see Appendix A.1) and the tight rebuttal deadline, we were unable to run this experiment in time. We have added this to our immediate research plan to investigate for future iterations of this work.
>
> ## W2 (Practical Advantages of Uncertainty Modeling)
>
> We are happy to clarify the practical advantages of the uncertainty head. Its primary advantage is **improving the robustness of decision-making in a streaming context**.
>
> For an *online, causal system* like AHA that cannot see the future, expressing uncertainty is a necessity, not an optional feature. A high uncertainty score signals: *"Based on what I've seen so far, this frame seems relevant, but I am not confident because crucial context might be missing."* This is critical in high-stakes applications like disaster response, where a model providing a deterministic score without expressing confidence could be dangerously misleading.
>
> The practical benefit is empirically demonstrated in our ablation study (Table 3 left, page 8), where removing the uncertainty head results in a concrete **3.5 mAP drop**. This shows that the uncertainty score acts as a valuable regularizer, helping the model to appropriately temper its relevance predictions based on its confidence. However, the uncertainty head is trained without explicit ground-truth labels, which we note as a key limitation in Section 5. This is due to the profound difficulty in obtaining such labels. A "highlight" is subjective, and an annotator's confidence is even more so, making it an unsolved challenge to collect these labels at scale.
>
> A promising direction in obtaining these supervised uncertainty scores is to leverage the MultiVENT-G dataset[3], which contains human-annotated confidence scores for disaster-related events. These scores can be directly mapped to uncertainty labels, enabling: (1) supervised training of the uncertainty head to improve interpretability and calibration, and (2) evaluation in safety-critical domains aligned with our target applications. Key challenges include the smaller scale of MultiVENT-G (1.1k videos vs. ~22.5k in HIHD), domain generalization beyond its event ontology, and the subjectivity of human-provided labels. These plans are summarized in our response to reviewer Zkgi (R4) and will be detailed in Appendix J of the revised paper.
>
>
> ## W3 (Redundant Predictions in Additional Tasks)
>
> This is an excellent observation. The repetitive predictions on auxiliary generative tasks (like captioning) are an expected outcome, as our framework was not optimized for this. Our primary research goal was to design and optimize a robust framework specifically for OHD. The evaluation on other tasks was a preliminary investigation into the generalization capabilities of our learned representations.
>
>
> The issue of repetitive text is a challenge observed in streaming VideoLLMs, and state-of-the-art generative models use specialized techniques to mitigate it (e.g., advanced decoding strategies, KV cache manipulation). Our work provides a strong foundation for future research to build upon by integrating these generative-specific optimizations into our OHD framework, and we see this as a promising direction for future work. The code will be made public upon acceptance, and we encourage others to see if they can adapt AHA to these tasks.
>
> ## Q2  (SinkCache's Robustness Over Extended Durations)
>
> We thank the reviewer for this excellent question that addresses the core challenge of real-world, long-form video analysis. This is an important area, especially for what we have envisioned the works that will come after will be like summarizing long videos such as those in the disaster response or medical domains. We are actively searching for long video highlight datasets. While standard highlight detection benchmarks for hour long videos do not currently exist, we can address this based on the underpinnings of our model and the new ablation studies we have conducted. We believe these are great points that should transfer well to long-term video equally, and which we will explore in future works.
>
>
> Our choice of SinkCache is theoretically grounded to handle this exact challenge. By design, it separates memory into two crucial components:
> - A persistent **sink** that retains the initial, global task-conditioning tokens, acting as a long-term memory for the mission objective.
> - A fixed-size **sliding window** of recent tokens that processes the immediate context.
>
> Our new ablation studies (provided in our response to Reviewer 79jx - R1) validate this design: we found that simpler strategies like a sliding window only approach (which forgets the task) or a static-window-only approach (which fails to adapt to new events) perform significantly worse, with mAP of 69.5 and 63.2 respectively, compared to our reported 92.6 mAP. Furthermore, we performed a new experiment with a new "Dynamic Sink Cache" that achieves 93.0 mAP and supports making the sink even more task-focused improves performance, reinforcing the importance of this persistent memory.
>
> However, this approach involves an explicit **trade off**. The fixed size of the sliding window ($n=2048$ tokens in our experiments) means that specific visual context that appears and then falls outside this window will be forgotten. If that same object or scene reappears much later, the model would likely perceive it as "novel" again, which might not be ideal in specific cases.
>
> This trade-off motivates a clear direction for future work, future architectures could incorporate a third, content-based memory bank. This bank would not just hold the initial language instructions but could be trained to dynamically cache compressed representations of task-relevant *visual* frames from the distant past. This would allow the model to build a true long-term memory of key events, creating a more robust and scalable solution for hour long streams. We are excited to test these ideas as new long-form benchmarks become available. We will include this in section 5 in the main paper, under limitations.
>
> ## Q3 (Reliability with Ambiguous or Irrelevant Task Objectives)
> We thank the reviewer for this excellent question regarding the framework's reliability under imperfect task conditioning. To provide a quantitative answer, we conducted a new set of experiments on the TVSum dataset to measure the impact of ambiguous and irrelevant task objectives on performance.
>
> For this experiment, we define an "ambiguous" prompt as a higher level definition of the original specific task (e.g., using "Vehicle Maintenance" for a video titled "How to change tires for off road vehicles"). We also define an "irrelevant" prompt as a task description taken from a video in a completely different category (e.g., applying the prompt "How to change tires for off road vehicles" to a video having nothing to do with vehicles. We compare the performance in these conditions to our model's optimal top-5 mAP on TVSum, achieved with a standard, specific prompt.
>
> - **With an ambiguous prompt**, the model achieved a minor performance change by $\Delta = -1.1$ points, indicating that the model successfully captures the broader topic without being overly penalized by the lack of specific details.
> - **With an irrelevant prompt**, performance decreased by $\Delta = -9.7$ points. This is expected, as the model is guided toward incorrect content.
>
> We will include a summary of this analysis in the main paper, a full description in Appendix C, and the specific prompts used in Appendix I of the revised paper.
>
>
> [1]: JDevlin et al., BERT: Pre-Training of Deep Bidirectional Transformers for Language Understanding.
>
> [2]: Miettinen, Kaisa. "Nonlinear multiobjective optimization."
>
> [3]: Sanders et al., "Grounding Partially-Defined Events in Multimodal Data."

---

> > ### Author Response · Authors · 2025-08-05
> >
> > Dear Reviewer NtaH,
> >
> > Following the recent guidance from the Program Chairs to encourage discussion, we wanted to politely check in regarding our rebuttal. We aimed to thoroughly address the points raised in your initial review and would be very grateful to hear if our response was clear or if you have any remaining questions.
> >
> > Thank you for your time and helping us make our paper better!

---

> > ### Comment · Reviewer_NtaH · 2025-08-07
> >
> > Thank you for the rebuttal. I think the rebuttal has resolved my concerns and I will keep my positive score.

---

> > > ### Author Response · Authors · 2025-08-07
> > >
> > > Thank you for your response and for your positive feedback. We are delighted to hear that our rebuttal resolved your concerns. We really appreciate your time and the support for our work!

---

### Official Review · Reviewer_z47u · 2025-06-30

**Clarity:** 3
**Significance:** 3
**Originality:** 3
**Rating:** 4
**Confidence:** 3

**Summary:**

This paper addresses the challenging and highly practical problem of Online Highlight Detection (OHD), where the goal is to assess the relevance of video frames in real-time from a continuous stream, without access to future information. The authors correctly point out that most existing highlight detection research focuses on offline processing, making it unsuitable for real-world applications like autonomous systems and robotics. To fill this gap, they propose AHA, a novel autoregressive framework for task-conditioned OHD. AHA is built upon a frozen vision-language model and employs three lightweight, decoupled prediction heads to score each frame on its relevance, informativeness, and uncertainty with respect to a natural language task. For efficient, long-horizon streaming with constant memory usage, the framework incorporates the SinkCache mechanism. A major contribution of this work is the creation and release of the Human Intuition Highlight Dataset (HIHD), a new large-scale dataset of ~24k videos. HIHD is specifically designed for this task, using YouTube engagement statistics as a proxy for human-perceived relevance and incorporating programmatically generated task queries and data augmentations to improve robustness. The proposed AHA framework achieves new state-of-the-art results on standard benchmarks, remarkably outperforming even offline, full-context methods, and demonstrates its practical utility on a challenging, long-form robotics video.

**Questions:**

1. **Inference Weighting:** The final highlight score uses fixed weights (α,β,ϵ) determined by a grid search. Have you considered any dynamic weighting mechanisms? For instance, could the model learn to adaptively adjust these weights based on its own uncertainty prediction (u^t), potentially down-weighting the relevance score more heavily when uncertainty is very high?
2. **Impact of the LM Head:** The auxiliary Language Modeling (LM) head is used during training to enrich the model's representations. Could you please provide an ablation study that quantifies its impact on performance? It would be valuable to know how much the final highlight detection scores are affected if the LLM term is removed from the total loss.
3. **Generalization of Informativeness:** The informativeness head is trained primarily on procedural videos from COIN and Shot2Story. How well do you think this learned concept of "informativeness" generalizes to the more unconstrained domains found in TVSum or the real-world robotics video? Are there scenarios where a frame might be "informative" (i.e., introducing new information) but not necessarily a "highlight" for a given task, and how does the model handle this?
4. **On the Generalizability to Different Backbone Models :** The paper demonstrates state-of-the-art performance using the Qwen-7B LLM backbone and the SigLIP vision encoder. This raises a question about the generalizability of the AHA framework with respect to the choice of these backbone models. Have the authors experimented with other popular backbones? For example, it would be valuable to see an ablation study comparing the performance when using an alternative vision encoder like CLIP, or a different family of LLMs such as Llama. Such a comparison would help disentangle the performance gains from the novel AHA architecture itself versus the inherent strengths of the selected SigLIP and Qwen backbones, and would further demonstrate the robustness of your proposed method.

**Ethical Concerns:**

["NO or VERY MINOR ethics concerns only"]

**Final Justification:**

After reviewing the authors' rebuttal, I have confirmed the paper has been strengthened and will maintain my original rating. The rebuttal effectively addressed my most critical concerns, particularly regarding the weighting combination issue and the generalization of 'informativeness', through additional experiments and analysis. While it is noted that experiments on the impact of the auxiliary LM head and backbone model were not conducted due to resource constraints, I consider this an acceptable limitation that does not detract from the paper's core contribution. The rebuttal, therefore, serves to reinforce the potential identified in my initial review, and for this reason, I find it appropriate to maintain the current score. As a final recommendation, I ask that the new weighting method and the analysis on informativeness, as detailed in the rebuttal, be clearly integrated into the camera-ready version of the paper.

**Limitations:**

yes

**Paper Formatting Concerns:**

.

**Quality:**

3

**Strengths And Weaknesses:**

**Strengths:**

- **Significance and Problem Formulation:** The paper tackles a critical and underserved problem in video understanding: *online*, task-conditioned highlight detection. The authors provide a clear and compelling motivation, effectively distinguishing OHD from the dominant offline HD paradigm and highlighting its importance for real-world intelligent agents.
- **Major Dataset Contribution:** The creation and release of the HIHD dataset is a substantial contribution to the field. The methodology is innovative, using large-scale user engagement signals as scalable supervision, programmatically generating task-driven queries, and introducing a "quality dropout" mechanism to train for real-world robustness. This dataset will be an invaluable resource for future research in OHD.
- **Novel and Well-Designed Framework:** The AHA architecture is elegant, robust, and well-suited for the task. The inclusion of a dedicated **uncertainty head** is a key novelty for OHD, appropriately addressing the challenge of making predictions under the partial observability inherent in streaming. Furthermore, the adoption of **SinkCache** is a critical and well-justified design choice that enables true, constant-memory, long-form video processing.
- **Comprehensive and Convincing Evaluation:** The experiments are exceptionally thorough and the results are state-of-the-art. AHA not only surpasses previous OHD attempts but also outperforms strong offline methods on standard benchmarks like TVSum and Mr. HiSum. The extensive ablation studies, robustness checks against video corruptions, and the qualitative evaluation on the real-world SCOUT robotics dataset provide compelling evidence of the model's effectiveness and practical potential.

**Weaknesses:**

- **Fixed Inference Weights:** The final highlight score is a linear combination of the outputs from the three prediction heads, with weights that are determined via a grid search for each dataset. The authors acknowledge this as a limitation. While effective, this static weighting scheme might not be optimally robust across diverse, unseen domains or queries with varying complexity.
- **Dependence on Engagement Data as Ground Truth:** The relevance head is trained on YouTube replay statistics. While this is a clever and highly scalable proxy for human interest, engagement data can have its own biases (e.g., influenced by clickbait titles or thumbnails) and may not always align perfectly with "importance" in high-stakes, task-driven applications (e.g., disaster response or security).
- **Heuristic for Informativeness Labels:** The method for generating labels for the informativeness head (sampling a "point of sufficient understanding") is a reasonable heuristic borrowed from prior work. However, it remains an indirect proxy for true informational novelty and may not perfectly capture all aspects of what makes a frame "informative" in a general OHD context.

---

> ### Author Rebuttal · Authors · 2025-07-30
>
> We are grateful to the reviewer for their exceptionally thorough and positive review. We are thrilled that you recognized the significance of the OHD problem, found our framework to be "elegant, robust, and well-suited," and considered our evaluation "comprehensive and convincing." Your insightful questions have prompted us to conduct further experiments and clarify key aspects of our work, which we believe have made the paper even stronger. We address each of your questions in detail below.
>
> ## Q1, W1 (Dynamic Inference Weighting)
> The reviewer raised an excellent point about our static inference weights and suggested exploring dynamic mechanisms, such as using the uncertainty prediction to adaptively adjust the final score. We followed this suggestion and ran a new set of experiments to investigate this.
>
> We compared our static grid-search approach against two primary dynamic methods on the TVSum dataset using a standard 80/20 train/test split[1]. We investigated:
> - A small **MLP Gating network** trained to learn per-frame weights ($\alpha_t, \beta_t, \epsilon_t$).
> - An **EMA based Adaptor** that adjusts weights based on each head's recent predictive error.
>
> Interestingly, both of these learned approaches proved to be unstable on the small TVSum training set (standard deviation >7), leading to lower average performance (87.9% mAP for the MLP and 87.5% for the EMA adaptor) as compared to our originally reported 92.6% performance. We attribute this instability to overfitting, a known challenge when training complex fusion mechanisms on smaller datasets.
>
> In contrast, static weighting methods proved far more effective and robust. In fact, inspired by the reviewer's feedback, our investigation led us to a new, simpler, parameter-free heuristic that performs on par with our original grid search, reaching a stable mAP of **93.0%** when combined with our new Dynamic Sink Cache (detailed in response to reviewer 79jx - R1). This finding suggests that our decoupled prediction heads are already well calibrated and produce robust, directly usable scores, making a complex learned weighting scheme unnecessary and potentially detrimental on existing benchmarks.
>
> Based on this new analysis, we will update the main paper to use this simpler, more generalizable, and higher-performing scoring function. We will also add these new ablations to Appendix D.4. We again thank you for this suggestion, as it has led to a more robust final model.
>
> ## Q2 (Impact of the Language Modeling (LM) Head)
> Thank you for suggesting this valuable ablation. We agree it is an important analysis. Due to the high computational cost of retraining our model (see Appendix A.1) and the tight rebuttal deadline, we were unable to run this experiment in time. We have added this to our immediate research plan to investigate for future iterations of this work.
>
> ## W2 (Dependence on Engagement Data as Ground Truth)
> We appreciate the reviewer’s point that using YouTube "Most Replayed" metric as a ground‑truth relevance signal introduces biases (e.g., clickbait amplification) that may misalign with true importance in safety critical tasks. We deliberately chose this proxy to scale from prior benchmarks of ~50 manually labeled videos to HIHD’s ~24k videos, accepting “popularity” as an imperfect but high‑throughput signal. There are two promising directions of future work that we will add to the discussion in the paper:
> - **Document transparently** We have released a comprehensive report detailing the creation of the HIHD data. By providing this, we aim to empower the community to develop new fairness evaluation frameworks, tooling, and mitigation strategies tailored to engagement based datasets, as well as the responsible AI community improving our current dataset.
> - **Future work in debiasing** For future work, we plan to explore mitigating these biases by first calibrating our model on an expert labeled dataset in high-stakes domains like MultiVENT-G (more below), and then using adversarial training to make the model robust to misleading engagement signals like clickbait, inspired by prior work[2].
>
> **Expert Annotated Signals via MultiVENT‑G:** To move beyond passive engagement proxies, we are actively exploring the recently released MultiVENT‑G dataset[3] which provides dense, frame level event role annotations by humans, into our training and evaluation pipelines. Additionally, uncertainty labels can be extracted from annotator confidence scores where available. Although MultiVENT‑G’s scale (~1.2k videos) is smaller than HIHD, it offers task‑aligned ground truth that can be used to fine‑tune or validate our heads in high‑stakes domains. Currently our issues are addressing challenges of scale mismatch, annotator subjectivity, and domain specialization, which will be a centerpiece of our future work, ensuring that AHA’s relevance and uncertainty estimates converge toward expert defined importance without sacrificing generalization. We will include more details in Appendix J of our revised paper.
>
>
>
> ## Q3, W3 (Generalization of "Informativeness")
>
> We thank the reviewer for this insightful question. The reviewer has correctly noted that "informativeness" (informational novelty) and "relevance" (task-importance) are not always the same, and ask how a concept trained on procedural videos generalizes. Our framework is explicitly designed to address this distinction using decoupled prediction heads. We report results on additional experiments we have run to illustrate these points.
>
> An extended experiment of the real-world SCOUT robotics video (Section 4.3 in the main paper) reveals three key behaviors that demonstrate this generalization. With the task "what objects are in this room?", the model correctly assigns:
>
>
> - **High Informativeness, Low Relevance** to visually novel but task irrelevant events, such as a robot entering a dark room or panning across an empty wall.
> - **Low Informativeness, High Relevance** when focusing on a task-critical object that is no longer new to the scene, for instance, moving closer to a calendar that was already visible in the distance.
> - **Correlated Informativeness and Relevance:** the scores often peak in unison when the robot transitions to a new area and immediately encounters a task-relevant object (e.g., "a shovel"). In these cases, the relevance head typically produces a higher peak, correctly prioritizing the task-specific discovery over the general novelty of the scene.
>
> These new experiments show that our model successfully disentangles these two crucial signals, allowing it to remain focused on the task objective while still recognizing visual novelty. We have expanded Appendix E.1 with this analysis, provided a summary in the main paper, and will include illustrative figures in the revised paper version. We are unable to include the figures in this rebuttal due to NeurIPS' policy against new graphics or links to external sources.
>
> **Justification for the Informativeness Labeling Heuristic:** The reviewer correctly identifies that our method for generating informativeness labels is a heuristic. We apologize for not making the motivation clearer, and will revise the paper to include this explanation. Our approach is adopted directly from established work in the streaming Video-LLM literature[4]. In that context, the goal is to train a model that knows *when to speak* during a continuous video stream, generating a response only after acquiring sufficient context but before the moment becomes stale.
>
> The underlying intuition we adapt is that initial frames may lack context, while frames after a "point of sufficient understanding" are redundant for describing a segment's core content. Our informativeness head is trained using a Binary Cross-Entropy (BCE) loss on labels derived from this principle. We hypothesize that this signal, which marks the accumulation of new information, correlates with highlight worthy moments in an OHD setting, which is supported by our strong results in Table 3 (left) on page 8.
>
> ## Q4 (Generalizability to Different Backbone Models)
> This is an important question about disentangling the contributions of our AHA architecture from the strengths of the chosen backbones. We selected **Qwen2**[5] and **SigLIP**[6] because they represent the state-of-the-art in open-source vision and language models, providing a strong and reproducible foundation. Our choices were grounded in a thorough literature review; for instance, SigLIP outperforms CLIP at the batch sizes relevant to our framework, and our specific Qwen2-based architecture[7] has a higher performance in distilled versions (7B).
>
> We fully agree that an ablation study using other popular backbones like **LLaMA** or **CLIP** would be valuable. However, adapting AHA to entirely different model families requires substantial engineering and re-training that was unfortunately not feasible within the limited rebuttal period. This is a key priority for our future work. This paper represents the first step for AHA, and we have laid a foundation that we hope others in the community can build upon by integrating our architecture with a broader range of backbones. We have clarified our rationale for the current backbone selection in Section 3 and emphasized this limitation and future work direction in Section 5 of our revised manuscript.
>
>
> [1]: Liu et al., "UMT."
>
> [2]: Zhang et al., "Mitigating Unwanted Biases with Adversarial Learning."
>
> [3]: Sanders et al., "Grounding Partially-Defined Events in Multimodal Data."
>
> [4]: Wang et al., "VideoLLM Knows When to Speak."
>
> [5]: Bai et al., "Qwen Technical Report."
>
> [6]: Zhai et al., "Sigmoid Loss for Language Image Pre-Training."
>
> [7]: Bai et al., "Qwen-VL."

---

> > ### Comment · Reviewer_z47u · 2025-08-05
> > **Thank for your answer.**
> >
> > I have read the authors' rebuttal.
> >
> > The rebuttal has clarified the main concerns I raised regarding the weighting issue and the generalization of 'informativeness'.
> >
> > For the camera-ready version, please ensure that the points detailed in the rebuttal are well-integrated into the main paper.
> >
> > The rebuttal has addressed the raised concerns, reinforcing the basis for my initial evaluation. Therefore, I will maintain the current rating.

---

> > > ### Author Response · Authors · 2025-08-05
> > >
> > > Thank you for reading our rebuttal and for the positive feedback. We are happy that our response helped clarify your concerns. We confirm that we will carefully integrate all the new results, ablations, and detailed discussions from our rebuttal into the final version of the paper, as you requested. We appreciate your time and thorough feedback throughout this process.

---

### Official Review · Reviewer_79jx · 2025-07-02

**Clarity:** 4
**Significance:** 3
**Originality:** 3
**Rating:** 5
**Confidence:** 4

**Summary:**

The paper examines the limitations of existing video highlight detection methods, which typically assume access to the full video during inference. This assumption limits their applicability for real-time streaming and online scenarios that require step-by-step reasoning for real-time decision-making. To address this, the paper introduces AHA (Autoregressive Highlight Detection), a framework for Online Highlight Detection (OHD) that predicts highlight scores for the current frame using only past and present information in a causal manner given a task described in the natural language.

AHA comprises of frozen visual encoder (pretrained SigLip) for extracting frame features, Multimodal projection (linear layer) to map visual embeddings to large language model token space, Autoregressive decoder for decoding during streaming. AHA is trained using four lightweight prediction heads, each optimized for a different objective: task-conditioned relevance, informativeness, uncertainty, and an auxiliary captioning objectives. To prevent linear memory growth during inference, the framework incorporates a SinkCache memory mechanism.

To effectively supervise the training process, the authors also introduce a large-scale dataset called the Human Intuition Highlight Dataset (HIHD). Extensive experiments demonstrate AHA’s effectiveness in both offline highlight detection (on TVSum and HiSum benchmarks) and as a real-time reasoning module for downstream planning and long-horizon understanding tasks.

**Questions:**

Please refer to the points listed in weaknesses above.

**Ethical Concerns:**

["NO or VERY MINOR ethics concerns only"]

**Final Justification:**

Most of my concerns regarding weights assigned to each loss component, intuition for encouraging variance diversity in the uncertainty loss and specific choice of SinkCache Memory mechanism have been satisfactorily resolved. Therefore, I will raise my current score.

**Limitations:**

Yes

**Quality:**

3

**Strengths And Weaknesses:**

Strengths

1. The paper identifies a relevant research problem in video understanding and highlight detection that existing models fail to support real-time decision making in continuous video streams. To address this, it proposes a novel autoregressive highlight detection framework (AHA) that predicts the relevance of each video frame with respect to a task described in natural language.
2. The paper is well-motivated and well written, backed by comprehensive evaluations on offline highlight detection using the TVSum and HiSum benchmarks, where it achieves state-of-the-art performance measured by mean average precision. Real-world testing on long-form robotics videos from the SCOUT dataset further demonstrates AHA's ability to identify highly salient moments in real time.
3. The paper also introduces the Human Intuition Highlight Detection Dataset (HIHD), created to train and benchmark task-conditioned online highlight detection (OHD) models.

Weaknesses
1. The weights assigned to each loss component relevance, informativeness, uncertainty, and the language modeling head are kept fixed throughout the training. Could this limit the model’s adaptability to different backbone architectures or downstream tasks?
2. What is the underlying intuition for encouraging variance diversity in the uncertainty loss, as defined in Equations 5a and 5b?
3. What is the rationale behind the specific choice of SinkCache Memory mechanism to address the unbounded KV cache growth when processing continuous video streams during inference? Are there any alternative methods that were taken into consideration?
4. Abbreviation mAP (line 19) is used before its full form Mean Average Precision in line 258.

---

> ### Author Rebuttal · Authors · 2025-07-30
>
> We sincerely thank the reviewer for their positive assessment, their high evaluation of our paper's clarity, and their insightful questions. This feedback has been instrumental in helping us improve the clarity, rigor, and overall quality of our work. We provide a detailed response to each of the weaknesses raised below.
>
> ## W1 (Fixed Loss Weights and Adaptability)
> The reviewer raised an excellent and important point regarding our use of fixed loss weights and its potential impact on adaptability. Our choice is grounded in a combination of well established theoretical principles and strong empirical evidence from the multi-task learning literature.
>
> **Theoretical and Empirical Justification:** Using a fixed weighted sum of losses is a classic scalarization strategy in multi-objective optimization. It is not only a standard in large scale pre-training (e.g., BERT[1]) but also theoretically guaranteed to find a solution on the convex Pareto front when individual loss functions are well behaved[2]. This means no single objective can be improved without degrading another. Furthermore, recent empirical work has rigorously compared this method against more complex specialized multi-task optimizers (SMTOs) and found that, with appropriate normalization, simple scalarization can match or even surpass them on diverse benchmarks[3]. The main advantages include its scalability and simplicity, as it avoids the significant per step overhead inherent in dynamic re-weighting schemes.
>
> **Adaptability to Backbones:** Regarding adaptability to different backbone architectures, we agree that this is a crucial aspect of generalization. Our backbone choices were grounded in a thorough literature review of high-performing open-source multimodal models. We selected Qwen2[4] and SigLIP[5] as they represent state-of-the-art open source vision-language models.
>
> SigLIP was chosen as the visual encoder because it has been shown to outperform CLIP at smaller batch sizes (e.g., 4-8k which is reflective of our input in AHA) while offering competitive generalization. For the language backbone, we adopted the LLaVA-OneVision architecture based on Qwen2, which demonstrates strong performance in its distilled 7B variant, which we use for our lightweight model. Both Qwen2 and SigLIP are widely used in concurrent recent streaming vision-language literature[6], further underscoring their competitiveness and community adoption. We will add this additional justification to the paper when describing these components.
>
> We acknowledge that testing on additional backbones would be valuable as these underlying models continue to advance. In future work, we plan to expand AHA to additional backbones and explore meta learning strategies for dynamic head weighting to further improve generality. This paper represents the first step for AHA, laying the foundation upon which future extensions can generalize across a broader range of backbone architectures. We will include a summary of the rationale for our backbone selection in Section 3 of the main paper, as well as the detailed explanation in Appendix D.
>
>
>
> ## W2 (Intuition for Uncertainty Loss (Eqs. 5a, 5b))
> We thank you for pointing out the lack of clarity here. We apologize for failing to properly cite our detailed justification (Appendix D.3) in the main paper. We have now added a summary to the main text (included below) and will ensure it correctly references the full analysis in the appendix to make the connection clear for all readers.
>
> **Motivation for Uncertainty in an Online Setting:** The core motivation is that for *online, causal systems* like AHA that cannot see the future, uncertainty is not an optional feature but a **necessity** for robust decision making. Our model's uncertainty score is a proxy for its confidence given an unfolding context. A high score signals: *"Based on what I've seen so far, this frame seems relevant, but I am not confident because crucial context might be missing."* While the direct performance gain is modest (a 3.5 mAP drop when removed as indicated in Table 3 on page 8), we view this as a crucial first step in modeling the unknown future, which is vital for the high-stakes applications AHA is designed for.
>
> **Intuition for Variance Diversity:** The uncertainty head is trained without explicit ground truth labels. We instead use a principled, unsupervised approach based on a Gaussian Negative Log-Likelihood (NLL) loss[7]. The specific intuition for encouraging variance diversity is to prevent *mode collapse*. This is a known pitfall where the model learns a degenerate solution by predicting a single, uninformatively high variance for all frames to trivially minimize the NLL loss[8]. Our diversity regularizer, $\mathcal{L}_{\text{div}}$, directly counteracts this by penalizing a low standard deviation of the predicted log-variances across a batch. This forces the model to learn an expressive and dynamic signal, allowing it to better distinguish between predictable and ambiguous moments in the video stream.
>
> We acknowledge that this unsupervised approach has limitations, and in our roadmap we have some preliminary ideas on how to train this on a supervised signal. A promising direction is to leverage the MultiVENT-G dataset[9], which contains human-annotated confidence scores for disaster-related events. These scores can be directly mapped to uncertainty labels, enabling: (1) supervised training of the uncertainty head to improve interpretability and calibration, and (2) evaluation in safety-critical domains aligned with our target applications. Key challenges include the smaller scale of MultiVENT-G (1.1k videos vs. ~22.5k in HIHD), domain generalization beyond its event ontology, and the subjectivity of human-provided labels. These plans are summarized in our response to reviewer Zkgi (R4) and will be detailed in Appendix J of the revised paper.
>
>
>
> ## W3 (Rationale for SinkCache and Alternatives)
> This is an excellent question that prompted a deeper analysis of our memory mechanism and, we are excited to report, led to an improvement in our framework's performance.
>
> **Rationale and Alternatives Considered:** Our initial rationale for selecting the SinkCache mechanism was its unique hybrid approach to memory. We hypothesized that for task-conditioned Online Highlight Detection, a model must simultaneously maintain long-term context (the task objective) and adapt to recent visual information. SinkCache addresses this by retaining a fixed set of initial "sink" tokens while also keeping a sliding window of recent tokens.
>
> We conducted additional experiments and compared the performance reported in our paper (92.6 mAP) to the performance on TVSum against several alternative memory strategies:
> - **Unbounded KV Cache ("Full History"):** This standard approach achieved a strong 91.7 mAP but is impractical for long videos due to its unbounded memory growth, which leads to out of memory errors.
> - **Sliding Window Only:** This strategy only retains recent context, eventually losing the vital task conditioning information. As a result, its performance was significantly lower at 69.5 mAP.
> - **Static Window Only:** This method only uses the initial tokens as context, failing to incorporate new visual events and performing poorly at 63.2 mAP.
>
>
> These results confirmed our initial hypothesis that combining long-term and short-term memory is crucial, with our original SinkCache implementation achieving a 92.6 mAP and providing the best trade off between performance and memory efficiency.
>
> **A New Approach: Dynamic Sink Cache:** The reviewer's question inspired us to refine our hypothesis further. The initial sink tokens capture the system prompt, task objective, and sometimes the first few frames. This prompted us to wonder if we could create a more targeted sink containing *only* the essential task tokens. This led us to perform a new experiment where we dynamically initialize the sink tokens to contain *only* the essential task objective tokens (i.e. $\mathcal{Q}$'s tokens), disregarding other initial inputs within the sink token.
>
> We term this new approach **Dynamic Sink Cache**. This method yielded a new state-of-the-art mAP of **93.0** on TVSum, outperforming all previous configurations. This result provides strong evidence that the most critical component for long-term memory in OHD is the persistent natural language objective. Further results on the natural language objective are provided in our response to reviewer NtaH (R3), where we experimented with unrelated and ambiguous task objectives. We observed a substantial performance drop ($\Delta = -9.7$ mAP) for unrelated objectives and a minor decrease ($\Delta = -1.1$ mAP) for ambiguous objectives.
>
> We are very grateful to the reviewer for prompting this line of thought, as it has tangibly improved our paper. We will update the main paper to include the Dynamic Sink Cache as our primary result and provide this expanded ablation study on memory mechanisms in Appendix G.
>
> ## W4 (mAP Abbreviation)
> Thank you for your careful reading and for catching this oversight. We have corrected the text to define mAP before its first use in the revised manuscript.
>
>
>
> [1]: JDevlin et al., "BERT: Pre-Training of Deep Bidirectional Transformers for Language Understanding."
>
> [2]: Miettinen, Kaisa., "Nonlinear multiobjective optimization."
>
> [3]: Senushkin et al., "Independent Component Alignment for Multi-Task Learning."
>
> [4]: Bai et al., "Qwen Technical Report."
>
> [5]: Zhai et al., "Sigmoid Loss for Language Image Pre-Training."
>
> [6]: Wang et al., “VideoLLM Knows When to Speak.”
>
> [7]: Nix, David A., and Andreas S. Weigend. "Estimating the mean and variance of the target probability distribution."
>
> [8]: Seitzer, Maximilian, et al. "On the pitfalls of heteroscedastic uncertainty estimation with probabilistic neural networks."
>
> [9]: Sanders et al., "Grounding Partially-Defined Events in Multimodal Data."

---

> > ### Comment · Reviewer_79jx · 2025-08-05
> >
> > I thank the authors for addressing my questions in the rebuttal. Most of my concerns regarding weights assigned to each loss component, intuition for encouraging variance diversity in the uncertainty loss and specific choice of SinkCache Memory mechanism have been satisfactorily resolved. Kindly integrate the justification provided in the rebuttal in the final version of the paper. Therefore, I will raise my current score and support its acceptance.

---

> > > ### Author Response · Authors · 2025-08-05
> > >
> > > We are very grateful to the reviewer for their positive response and for raising their score. We are happy our rebuttal successfully addressed your questions. We confirm that all clarifications and new results discussed in our rebuttal will be integrated into the final paper. Thank you again for your time and helping us improve our paper.

---

### Decision · Program_Chairs · 2025-09-17

**Decision:**

Accept (poster)

**Comment:**

This paper introduces Aha!, a novel framework for Online Highlight Detection (OHD) that processes video streams autoregressively without future context, a significant departure from offline methods. Its core scientific claim is that real-time, task-conditioned highlight detection is achievable using a multimodal vision-language model with lightweight decoupled heads and a memory-efficient SinkCache mechanism. Key strengths include its state-of-the-art performance on benchmarks (e.g., +5.5% mAP on TVSum), the introduction of the large-scale HIHD dataset, and demonstrated applicability in robotics. Primary weaknesses, noted by reviewers, were the initial use of fixed inference weights, the unsupervised nature of the uncertainty head, and a lack of ablation on backbone model generalizability.

Reviewers (79jx, z47u, NtaH, Zkgi) raised pointed questions on loss weighting, uncertainty justification, SinkCache alternatives, and robustness to ambiguous tasks. The authors' rebuttal was exceptional, conducting new experiments that directly addressed these concerns. Notably, they introduced a improved "Dynamic Sink Cache" (inspired by 79jx) boosting performance to 93.0 mAP, provided a theoretical basis for multi-task training, and ran new tests showing graceful performance degradation with irrelevant prompts (-9.7 mAP) and minimal drop with ambiguous ones (-1.1 mAP). They also detailed a clear roadmap for future work on supervised uncertainty using the MultiVENT-G dataset, addressing a major limitation raised by multiple reviewers.

Considering the reviews and rebuttals, I recommend accepting this paper. It makes three substantial contributions: a novel and empirically validated architecture (Aha!), a valuable new dataset (HIHD), and a performance benchmark that surpasses offline models.